



# A fjord dataset for multidisciplinary applications: Thirteen years of ocean observations in Sermilik Fjord, Southeast Greenland

Aurora Roth[1], Fiamma Straneo[2,1], James Holte[2,1], Margaret Lindeman[2], and Matthew Mazloff[1]

1. Scripps Institution of Oceanography, University of California at San Diego, La Jolla, CA, USA

2. Harvard University, Cambridge, MA, USA

**Correspondence:** Aurora Roth (a1roth@ucsd.edu) and Fiamma Straneo (fstraneo@seas.harvard.edu)

**Abstract.** As global atmosphere and ocean temperatures rise and the Greenland Ice Sheet loses mass, the glacial fjords of Kalallit Nunaat/Greenland play an increasingly critical role in our climate system. Fjords are pathways for freshwater from ice melt to reach the ocean and for deep, warm, nutrient–rich ocean waters to reach marine–terminating glaciers, supporting abundant local ecosystems that Greenlanders rely upon. Research in Greenland fjords has become more interdisciplinary and more observations are being collected in fjords than in previous decades. However, there are few long–term (> 10 years) datasets available for single fjords. Additionally, observations in fjords in general are often spatially and temporally disjointed, utilize multiple observing tools, and are rarely provided in formats that are easily used across disciplines or audiences. We address this issue by providing standardized, gridded summer season hydrographic sections for Sermilik Fjord in Southeast Greenland, from 2009–2023. Gridded data facilitate the analysis of coherent spatial patterns across the fjord domain, and are a more accessible and intuitive data product compared to discrete profiles. We combined ship–based conductivity, temperature, and depth (CTD) profiles with helicopter–deployed eXpendable CTD (XCTD) profiles from the ice mélange region to create objectively mapped (or optimally interpolated) along–fjord sections of conservative temperature and absolute salinity. From the gridded data, we derived a summer season climatological mean and root mean square deviation, summarizing typical fjord conditions and highlighting regions of variability. This information can be used by model and laboratory studies, fjord intercomparison studies, biological and ecosystem studies in the fjord, and provides context for interpreting previous work. All original profile observations, gridded data, and climatological products are publicly available in netCDF format at Arctic Data Center and GitHub. The code used has also been made available to facilitate continued updates to the Sermilik Fjord gridded data product and applications to other fjord systems.

## 1 Introduction

The glacial fjords of Kalaallit Nunaat/Greenland are key climate connectors– delivering freshwater (in liquid and solid form) to the ocean and warm ocean waters to the ice sheet. As global air and ocean temperatures rise and the Greenland Ice Sheet (GrIS)





melts at an accelerating rate, understanding fjord variability is critical to addressing large–scale questions of GrIS mass loss, freshening of the North Atlantic, and potential global ocean circulation changes (Straneo and Cenedese, 2015). Locally, fjords
are home to Greenlanders and their livelihoods are dependent on the future of fjords in our rapidly changing climate (Holm, 2010; Nuttall, 2020; Schiøtt et al., 2022). While abundant fjord ecosystems have been observed and utilized by Greenlandic peoples for thousands of years, there is increasing scientific interest for how physical fjord processes impact local ecosystems and biodiversity around Greenland and how these may evolve in a changing climate (Meire et al., 2017; Hopwood et al., 2020; Straneo et al., 2022).

Long–term and concurrent observations of atmosphere, glacier, ocean, and ecosystem variables at Greenland's coastal margins are essential for improving our understanding of glacial fjord systems. Motivated by science needs, the last decade has seen a significant increase in observations of all kinds collected in and near Greenland fjords. Notably, long–term (> 10 years) repeat oceanographic surveys have been carried out in Nuup Kangerlua (Godthåbsfjord) near Nuuk and in Young Sound in Northeast Greenland as part of the Greenland Ecosystem Monitoring MarineBasis Program (https://g-e-m.dk/gem-science-programme/
marinebasis-programme) and Greenland Institute of Natural Resources research campaigns (Juul-Pedersen, 2009; Mortensen et al., 2018). In Northwest Greenland, long–term oceanographic observations have been conducted in Kangerlussuaq (Inglefield Bredning) region near Qaanaaq (Sugiyama et al., 2020, 2025). Sermilik Fjord in Southeast Greenland, the focus of this study, has had nearly annual summer season oceanographic observations since 2008.

While many of the ice or atmospheric data at the margin of GrIS are available through remote sensing, reanalysis, or regional
climate model products (eg. Greenland Ice Sheet Mapping Project, ERA, RACMO), oceanographic data in fjords have been mostly collected by small research teams in isolated projects creating disparate datasets with widely varying characteristics distributed over many fjords (Schlegel and Gattuso, 2023). This makes it challenging to assemble these data in standard formats, typical of the large-scale oceanographic monitoring programs (eg. ARGO, GOSHIP), limiting their availability and usability.

Secondly, the environmental conditions and logistical constraints of working in fjords result in data being spatially and temporally disjointed. For example, repeat measurements at exact locations may be difficult to perform because of variable iceberg and sea ice presence and weather conditions. This makes it challenging to quantitatively compare different years or to provide modeling groups with mean properties instead of those based on a single survey. Even where repeat surveys exist, they may have been carried out using different instrumentation, sometimes within the same survey and/or by different groups
over the years. Given the growing interest in understanding Greenland's fjords, it is important to develop data protocols and repositories that standardize fjord data from multiple surveys of a single fjord, facilitate comparisons between data collected in different fjords, and provide boundary conditions, forcings, and comparisons for ocean and ice sheet models (Juul-Pedersen, 2009; Straneo et al., 2019; Schlegel and Gattuso, 2023).

Finally, as Greenland fjord research becomes increasingly interdisciplinary and collaborations with local Greenlandic com-
munities and government are being further strengthened, it is necessary that data be provided in formats that are accessible and usable by a wide range of users, including scientists from other disciplines (glaciologists, marine ecologists, social scientists), policy makers, educators, and local tourism operators.





Here, we present 1) a quality controlled hydrographic dataset from Sermilik Fjord in Southeast Greenland for 13 summer field campaigns occurring from 2009 – 2023, consisting of ship–based Conductivity, Temperature, and Depth (CTD) profiles and eXpendable Conductivity Temperature Depth (XCTD) profiles deployed from helicopters, 2) a standardized, along–fjord gridded dataset (Roth et al., 2025) combing both types of observations to increase usability by a diverse set of users, and 3) a climatological mean and root means square deviation (RMSD), calculated from the gridded dataset, that summarizes mean summertime fjord water properties and identifies regions with the greatest variability, which is of particular use for modeling studies.

Data collected include Conductivity, Temperature, and Depth (CTD) profiles from ships and eXpendable Conductivity Temperature Depth (XCTD) measurements typically deployed by helicopters in regions not accessible by ship. Some of these data have been described in previous studies (with data made available through data repositories) but the collective dataset and, importantly, the standardized gridded fields which allow for year–to–year intercomparison and for the derivation of a robust, climatological mean are new. Equally important, this study provides a procedure for the standardized gridding of the data, including an error estimate, where other observed variables in Sermilik Fjord (eg. dissolved oxygen, nutrients) can easily be gridded and incorporated into the database by multiple users. The method can also be easily adapted and used to build gridded datasets from profile observations in other fjord systems.

The annual gridded fields, created using an objective mapping method, facilitate the comparison of coherent spatial patterns between years, comparison to model output, comparison with other scattered observational variables, and for calculating fjord transport quantities. It also addresses the needs of interdisciplinary researchers, not familiar with CTD and XCTD data processing or treatment of observations from discrete profiles, but who are interested in the mean and variable properties of the fjord for studies related, for example, to glaciers and ecosystems.

From the annual gridded dataset, we calculated a summer climatology showing persistent hydrographic patterns of the system and regions of variability. The climatology is useful for informing model and laboratory studies, and provides context for interpreting previous studies in Sermilik Fjord.

The approach proposed here is a significant step toward creating a "glacial fjord data node"– a living data repository that standardizes long–term fjord observational records into a FAIR (Findable, Accessible, Interoperable, Reusable) data format that can facilitate interdisciplinary research.

## 2 Sermilik Fjord setting

### 2.1 Glaciological context

Sermilik Fjord, located in southeast Greenland (1), is long ( 90 km), narrow (5 km–10 km), and deep (550 m–900 m). The northern end of the fjord splits into three branching fjords with respective tidewater glaciers at the head of each branch– Helheim Gletsjer, Apuseerajik (Fenris Gletsjer), and Nigertiip Apusiia (Midgård, Midgaard, or Midgard Gletsjer) (Bjørk et al., 2015). Helheim, the largest of the three, is one of the largest and fastest flowing outlet glaciers of the GrIS. Due to Helheim's large volume of solid ice discharge, ( 30 Gt–38 Gt yr$^{-1}$ since 2000 from Mankoff et al. (2020)), freshwater input ( 500 m– 650

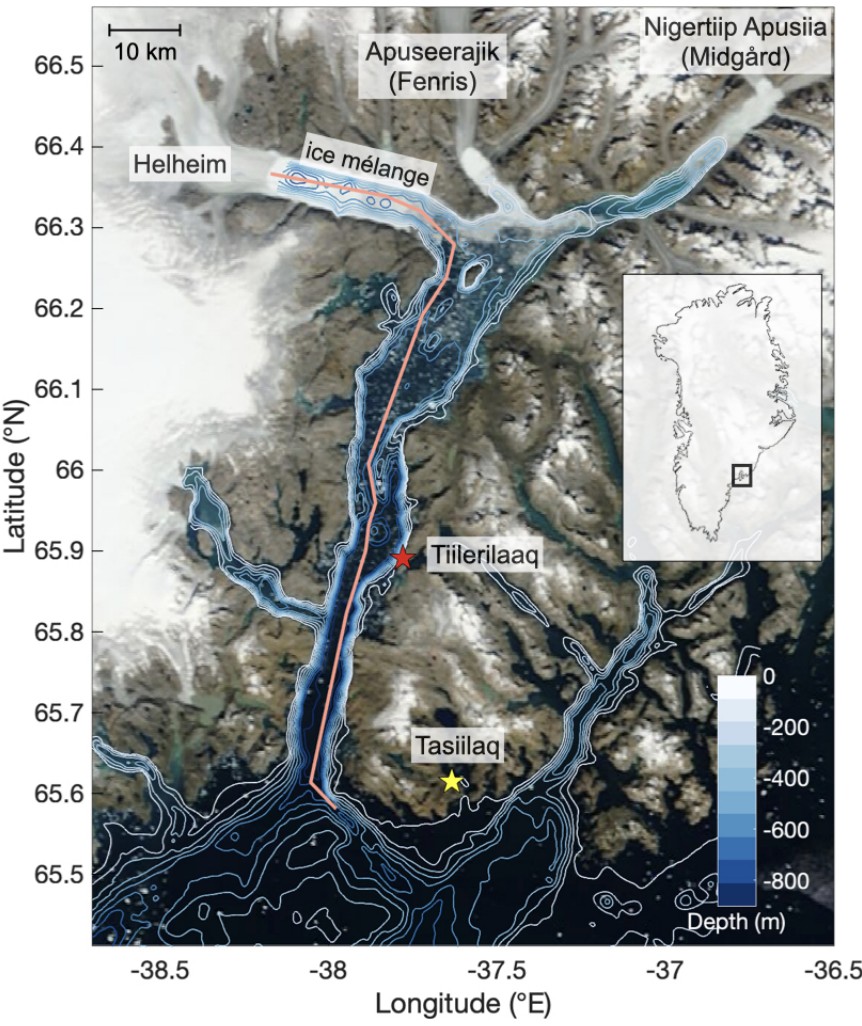

**Figure 1.** Map of Sermilik Fjord region in Southeast Greenland with major fjord branches and regions labeled. Bathymetry is shown as colored contours with 100 m increments and is derived from BedMachine Greenland v4 (Morlighem et al., 2017). The pink line represents the thalweg section of the fjord, used in plotting the bathymetry for the along–fjord sections (see text). The thalweg section end point is the Helheim Gletsjer terminus position in 2019. Background image is Terra MODIS corrected reflectance (true color) satellite image on August 12 2023. This image was obtained from NASA Worldview Snapshot application (https://worldview.earthdata.nasa.gov), part of Earth Science Data Information System (ESDIS).



m$^3$ s$^{-1}$ peak freshwater discharge in July from Mankoff et al. (2020)), and deep grounding depth ( 600 m) relative to the other glaciers, the physical dynamics of Sermilik Fjord are often studied by only considering the Helheim-Sermilik system. Since 2000, the terminus of Helheim has retreated 6 km and the glacier has lost an estimated 5 Gt–13 Gt yr$^{-1}$ of ice (Williams et al., 2021). Similarly, Apuseerajik (Fenris) and Nigertiip Apusiia (Midgård) have had consistently negative annual mass balance and terminus retreats ( 4 km and 11 km respectively) since 2000 (Williams et al., 2021; Huiban et al., 2024)).

## 2.2 Regional ocean context

The properties and circulation of the Sermilik Fjord region have been described in a number of earlier studies, and are briefly summarized below to provide context for this dataset.

The mouth of Sermilik Fjord opens onto the continental shelf where the East Greenland Coastal Current flows south carrying cold Polar Water (PW) (Conservative Temperature , < 0 °C, Absolute Salinity, 33.3 g kg$^{-1}$) of Arctic origin in the upper 200 m of the water column. Warmer, saltier Atlantic Water (AW) (> 3 °C, SA > 34.7 g kg$^{-1}$) from the Irminger Sea underlies the Polar Water on the continental shelf (Harden et al., 2014). A trough (400 m–900 m deep) extending from the fjord mouth across the continental shelf allows for the warm AW to be funneled from the shelf into the fjord (Straneo et al., 2010, 2011; Sutherland et al., 2014b; Snow et al., 2023). Shelf waters transported by the EGCC are known to enter Sermilik Fjord following the trough on the East side of the mouth, while there is strong preferential out fjord flow on the western side (Sutherland et al., 2014a).

## 2.3 Sermilik Fjord water masses

A relatively deep sill ($\sim$ 550 m) in Sermilik Fjord allows for direct exchange of PW and AW shelf waters. As a result, the same two layer structure can exist in the fjord with a pycnocline at 150–200 m depth. AW fills the deep regions along the entire length of Sermilik Fjord and is present at the terminus of Helheim Glestsjer leading to submarine melting (Straneo et al., 2011). Regional wind dynamics on the shelf, primarily in the winter months during storms, result in oscillatory changes to the depth of the shelf pycnocline relative to the fjord, resulting in intermediary flow and fjord–shelf exchange (Jackson et al., 2014, 2018).

In the summer months, the addition of subglacial discharge, submarine meltwater, and surface runoff creates more complex fjord circulation, hydrography and ice–ocean dynamics. At glacier termini, buoyant subglacial discharge plumes entrain and upwell the deep, warm AW leading to enhanced submarine melting at the terminus and creating a water mass referred to as Glacially Modified Water (GMW).In Sermilik Fjord, GMW appears as a relatively warm and salty intrusion in the upper 50 m–250 m of the water column (Straneo et al., 2011; Beaird et al., 2018; Lindeman et al., 2024). At the surface (< 50 m), the submarine meltwater of icebergs and the addition of surface runoff creates a fresh anomaly, referred to as surface GMW (sGMW) (Straneo et al., 2011; Lindeman et al., 2024).





## 2.4 Helheim Gletsjer ice mélange

Helheim Gletsjer has a perennial ice mélange consisting of icebergs and sea ice. The ice mélange region regularly extends up to 30 km from the Helheim terminus, however the total extent and area varies seasonally and interannually with changing glacier and calving dynamics (Foga, 2016). The submarine meltwater of icebergs in the mélange contributes to GMW and creates a cold temperature anomaly in the upper 100 m of the water column in the upper fjord area (Straneo et al., 2011; Enderlyn et al., 2016; Moon et al., 2018; Davison et al., 2022). Many questions remain about glacier–mélange–ocean feedbacks, in addition to questions about the role of subglacial discharge plumes in glacier–ocean dynamics at the terminus. Direct observations of these two fjord regions, ice mélange and subglacial discharge plume, are difficult and costly to obtain due to challenging ice conditions. The data presented here include observations from both regions for multiple summer seasons in Sermilik Fjord.

### 2.4.1 Sermilik Fjord Western science context

Sermilik Fjord became a site of intensive coordinated glaciological, atmospheric, and oceanic measurements starting in the late 2000s. Scientists aimed to understand the extent to which the ocean was playing a role in the retreat of Greenland's tidewater glaciers. At the time, there was little data from Greenland fjords and even high resolution ocean models did not resolve fjord–scale processes. Sermilik Fjord was chosen as a representative system of southeast Greenland glacial fjords because of the importance of Helheim Gletsjer to the dynamics of the GrIS as a whole (Straneo et al., 2016). More recently, Sermilik Fjord has been identified as a site for a Greenland Ice Sheet–Ocean Observing System (GrIOOS) due to the availability of interdisciplinary measurements previously collected there (Straneo et al., 2019).

Studies in Sermilik Fjord have greatly advanced our understanding of fjord systems and the role of fjord dynamics in connecting GrIS and the ocean. Important findings that have previously utilized portions of the CTD and XCTD hydrographic dataset presented here include 1) showing unequivocally that warm AW contacts glacier termini and drives submarine melting (Straneo et al., 2010, 2011, 2012), 2) demonstrating that glacial melt water entrains ambient fjord water and is exported out of fjords as GMW at depth (Straneo et al., 2011; Beaird et al., 2018), 3) teasing apart drivers of complex fjord circulation beyond traditional estuarine two layer circulation (Sciascia et al., 2013; Sutherland et al., 2014b; Jackson et al., 2014), 4) the importance of shelf processes for ice–ocean interactions in fjords (Jackson and Straneo, 2016; Spall et al., 2017; Sanchez et al., 2024; Snow et al., 2023), and 5) the role of icebergs and the ice melange in freshwater export and fjord properties (Enderlyn et al., 2016; Moon et al., 2018; Davison et al., 2022; Hughes, 2022). The works listed have included analyses of these hydrographic data with other observational platforms in Sermilik Fjord (eg. moorings) or are modeling studies using these hydrographic data for validation and/or forcing. Additionally, many advancements in our understanding of ice–ocean–climate processes as a whole have been made utilizing a wide range of datasets from the Helheim–Sermilik Fjord system from many different research groups.

More recently, the Sermilik Fjord region has been the site of studies addressing the relationships between physical ice and climate processes, fjord biogeochemistry, ecosystems, and local communities (Cape et al., 2019; Laidre et al., 2022; Straneo et al., 2022; Lindeman et al., 2024; Rathcke et al., 2025). Though specific project goals have varied over the years, CTD and





XCTD surveys have been reliably conducted almost every summer since 2009. This dataset is one of the longest oceanographic records of summer season water properties inside a southeast Greenland glacial fjord.

## 3 Data

### 3.1 CTD data

We present data from yearly summer surveys from 2009 to 2023, except for 2014 and 2020 (**??**). 364 shipboard CTD profiles and 71 XCTD profiles are included in this dataset. A variety of vessels and instrumentation have been used as methods, logistics, and instrument technology were improved and refined (1). From 2009–2013, conductivity, temperature, and pressure observations were collected with RBR XR–620 Titanium CTDs sampling at 6 Hz. Instrument accuracy is reported by the manufacturer as $\pm0.003$ mS/cm, $\pm0.002\circ$C, and $\pm0.05$of full depth scale for conductivity, temperature, and pressure sensors, respectively. Data from these CTDs required post–processing to correct unique pressure and conductivity offsets Straneo et al. (2010). Conductivity, temperature, and pressure were aligned prior to calculating salinity to account for the fact that the sensors are not physically co–located on the logger. Data were manually examined to address any salinity spikes or anomalous points.

Starting in 2015, a Sea–Bird SBE19plus CTD was used as the primary instrument, sampling at 16 Hz. Instrument accuracy is similar to the RBR XR–620 CTD and reported to be $\pm0.003$ mS cm$^{-1}$, $\pm0.002$ °C, and $\pm0.1$of full depth scale for conductivity, temperature, and pressure sensors, respectively. A RBR Concerto CTD was also mounted on the rosette and used for redundancy, with the same instrument accuracy as the RBR XR–620 and sampling at 16 Hz. SBE19plus CTD data were processed using Sea–Bird SBE data processing scripts to correct for lags between sensors, despike data, remove loops, and smooth data. Data were manually examined and any remaining anomalous points were removed.

All CTD profile data were vertically averaged to 1 m depth bins. We used the TEOS–10 Oceanographic Toolbox (McDougall and Barker, 2011) to convert in situ temperature to conservative temperature ($\Theta$), conductivity to absolute salinity ($S_A$), and pressure to depth. All profiles were smoothed with a low–pass boxcar filter. The complete dataset of processed CTD profiles, grouped by cruise, are available with all metadata at the Arctic Data Center (https://arcticdata.io/catalog/portals/sermilik/Data).

### 3.2 XCTD data

Starting in 2010, eXpendable Conductivity Temperature Depth (XCTD) probes were used in addition to shipboard CTDs (Straneo et al., 2011). The probes were deployed using helicopters to collect observations in the ice mélange and plume polynya regions where vessels cannot operate. Generally, the XCTD profiles are located in the near terminus region of Helheim Fjord, but additional XCTD profiles have been collected in GREENLANDIC NAME Midgaard Fjord and the main branch of Sermilik Fjord in more recent years (2).

XCTD instrument accuracy is reported as $\pm0.03$ mS cm$^{-1}$, $\pm0.02$ °C, and $\pm2.0$of full depth scale for conductivity, temperature, and depth respectively. The depth measurements are based on a constant fall speed and are thus less accurate than shipboard CTDs measuring pressure directly. All XCTD profiles were manually inspected and anomalous data points were



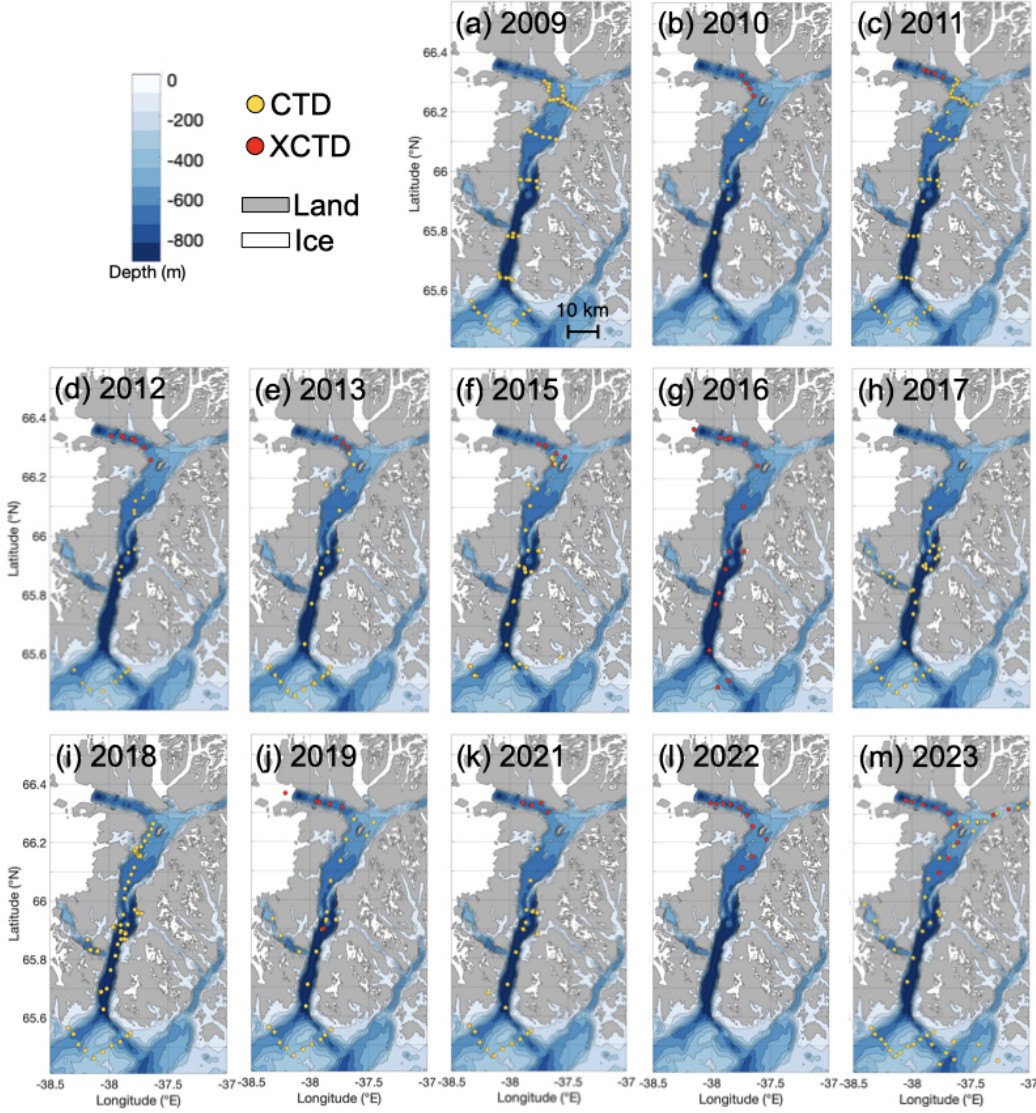

**Figure 2.** Locations of CTD (yellow dots) and XCTD (red dots) profiles for every summer survey in Sermilik Fjord included in this dataset. Bathymetry is shown as colored filled contours with 100 m increments. The bathymetry, land region, and ice regions are from BedMachine Greenland v4 (Morlighem et al., 2017). The land and ice regions correspond to outlines as they appear in the Greenland Ice Mapping Project (Howat et al., 2014) from the time periods 1999–2002 and 2013–2015. They are a static background for each year and do not represent specific ice extent and glacier terminus positions for a given year.





**Table 1.** Information about each shipboard CTD summer survey in Sermilik Fjord.

| Year | Survey Dates | Vessel | CTDs Used | Number of fjord CTD profiles | Number of shelf CTD profiles |
|------|-------------|--------|-----------|------------------------------|------------------------------|
| 2009 | Aug 19–25 | MY *Arctic Sunrise* | XR 620 Titanium RBR (s/n 18559) | 41 | 11 |
| 2010 | Aug 22–27 | *Pytur* | XR 620 Titanium RBR (s/n 18559) | 7 | 1 |
| 2011 | Aug 15–26 | *Viking Mads Alex* | XR 620 Titanium RBR (s/n 18559) | 39 | 12 |
| 2012 | Sept 14–20 | MV *Fox* | 2 x XR 620 Titanium RBR (s/n 17413, 18559) | 9 | 9 |
| 2013 | Aug 18–28 | *Viking Mads Alex* | 2 x XR 620 Titanium RBR (s/n 17413, 18559) | 13 | 14 |
| 2015 | Aug 2–11 | *Adolf Jensen* | SBEPLUS25 (s/n 251108); RBR Concerto (s/n 65584) | 22 | 8 |
| 2017 | July 15–22 | *Adolf Jensen* | SBEPLUS25 (s/n 251108); RBR Concerto (s/n 65584) | 21 | 10 |
| 2018 | Aug 3–14 | *Adolf Jensen* | SBEPLUS25 (s/n 251108); RBR Concerto (s/n 65584, 66129); RBR XR Titanium (s/n 18608) | 60 | 10 |
| 2019 | July 27–Aug 1 | *Adolf Jensen* | SBEPLUS25 (s/n 251108); RBR Concerto (s/n 66130) | 16 | 10 |
| 2021 | Aug 10–18 | *Adolf Jensen* | SBEPLUS25 (s/n 251108); RBR Concerto (s/n 66130) | 15 | 12 |
| 2023 | Aug 5–19 | RV *Tarajoq* | SBEPLUS25 (s/n 251108); RBR Concerto (s/n 66130) | 16 | 12 |

removed. We removed the top 4 m of each profile because it takes several seconds for the probe to equilibrate to the ocean
temperature once it enters the water and begins recording. The bottom of each profile was manually identified by spikes in the
conductivity measurements and cross–checked with expected bottom depth. The profiles were vertically averaged to 2 m depth
bins, in situ temperature and conductivity were converted to conservative temperature ($\Theta$) and absolute salinity ($SA$) using
TEOS–10 Gibbs–SeaWater Oceanographic Toolbox (McDougall and Barker, 2011), and a low–pass boxcar filter was applied
to each profile. These processed XCTD profiles are available for use and released with the same data and metadata format as
the CTD profiles at the Arctic Data Center (https://arcticdata.io/catalog/portals/sermilik/Data).

### 3.3 Combining CTD and XCTD data

XCTD measurements are less accurate compared to shipboard CTD measurements, and each XCTD profile uses a unique
probe. Therefore we must consider whether any differences in measured water properties from each instrument type are due





**Table 2.** Information about each XCTD survey in Sermilik Fjord. [*]One winter survey was conducted on March 15-16 2010 (Straneo et al., 2011). These profile data are available, but have not been gridded or included in the summer season climatology.

| Year | Survey Dates | Total number of XCTD profiles | Number of XCTD profiles in ice mélange |
|---|---|---|---|
| 2010[*] | Mar 15–16 | 5 | 1 |
| 2010 | Aug 26 | 4 | 4 |
| 2011 | Aug 26 | 4 | 4 |
| 2012 | Sept 14 | 7 | 7 |
| 2013 | Aug 22 | 3 | 3 |
| 2015 | July 27 | 5 | 5 |
| 2016 | Aug 9–11 | 16 | 6 |
| 2019 | July 31, Aug 6 | 7 | 6 |
| 2021 | Aug 11 | 4 | 4 |
| 2022 | Sept 11 | 10 | 6 |
| 2023 | July 12 | 12 | 5 |

to biases in individual XCTD probes rather than real property variability. Because AW properties below 400 m have the

195    smallest spatial and temporal variability throughout the fjord, we verified that the XCTD and CTD measurements for each year show matching AW properties within instrument error and known spatial variability, determined from the shipboard CTD measurements. No bias corrections were required for the XCTD data presented here, however this is not always the case and this verification step is critical when working with combined XCTD and CTD data.

### 3.4 Profile locations and timing

200    There is a wide variety of profile locations and timing of the surveys during the summer season (Tables 1, 2; Figs. 2, 3). This is influenced by different logistical constraints and priorities of each field campaign and the fjord conditions at the time. The fjord has a high concentration of icebergs making exact repetition of profiling locations difficult. In general, an attempt is made each year to sample along the centerline of the fjord following the deepest bathymetric path (thalweg section) over a continuous time period (Fig. 1). Note that in 2009 and 2011 across–fjord sections were performed. From this data, it was determined that across–

205    fjord variability is less significant compared to along–fjord variability, therefore surveys in the following years did not prioritize across–fjord sections (Straneo et al., 2011). Profiling locations are also influenced by recovering and deploying moorings, some of which are nearer to the coast than the fjord centerline. In 2018, the fjord survey was conducted in conjunction with specific iceberg surveys leading to many more profiles collected in the fjord that year. Other research priorities (eg. biogeochemical sampling) for a given year have influenced the final pattern of profile locations within the fjord.

210    XCTD profile locations in the ice mélange are limited to where gaps exist between icebergs and sea ice to deploy the probe. Notably, in 2016 and 2019, the Helheim subglacial discharge plume was visible at the ocean surface (known as a





plume polynya) and created ice–free openings (Melton et al., 2022). This allowed for the rare opportunity to deploy XCTD probes directly into the subglacial discharge plume waters at the glacier terminus, collecting observations of this undersampled and critical region. In 2016 and 2022 only helicopter–based XCTDs were used for the complete fjord survey due logistical constraints.

In contrast, the sampling locations on the shelf are repeated at nearly the same locations each year because icebergs are less present on the shelf. The V–shape configuration is designed to cross the trough twice, observing shelf waters flowing into the fjord using the East section and fjord waters flowing onto the shelf using the West section. All shelf profiles have been processed as described in Section 3.1 and are available with their associated fjord profiles for each year.

The surveys have occurred at different time periods in the summer (3). The seasonal dynamics of the fjord evolve over time and summer conditions can vary widely. Subglacial discharge and surface runoff begin to enter the fjord in June after the onset of the surface melt season. Both forms of freshwater entering the fjord peak in volume flux in late July and are usually negligible by late October (Mankoff et al., 2020). Buoyancy–driven circulation takes some time to set up with the addition of subglacial discharge. The continued addition of subglacial discharge throughout the summer can change the overall fjord stratification, altering the neutral buoyancy depth of the subglacial discharge plume over a summer season (Sanchez et al., 2023). Each year the surveys are capturing different time periods in this overall seasonal evolution of the fjord. Additionally, wind events on the shelf can drive shorter timescale (days to weeks) intermediary circulation on top of the buoyancy–driven circulation (Jackson et al., 2014, 2018). These events can quickly change fjord water mass properties over the timescale of a single fjord survey. Understanding the field campaign timing relative to freshwater inputs, wind conditions, regional climate conditions, is critical context for interpreting the data presented here.

# 4 Methods

## 4.1 Profile selection for analysis

Constructing along–fjord sections for each year required careful manual selection of the profiles that were collected nearest to the thalweg section (Figure 1) and capturing similar fjord conditions within a certain time period. If profiles in similar locations exist, but were collected at different time periods during the survey (eg. collected while sailing upfjord and then downfjord several days later), we only retained the profile that creates the best continuous synoptic section. If multiple profiles were collected in the across–fjord direction at the same along–fjord distance then those across-fjord profiles were averaged and the mean profile was used in the along–fjord section. Profiles near the shallower (< 300 m) fjord sides were not included. The final selection of individual profiles and mean profiles (averaged in the across-fjord direction) making up the best synoptic along–fjord section were then used as input data to create the gridded along–fjord data products.

Data from inside the plume regions from years 2016 and 2019 are not included in the along–fjord sections. This is because the gradients of properties are at a finer scale in this dynamic region than we are accounting for in the objective mapping process. These plume data are discussed separately.

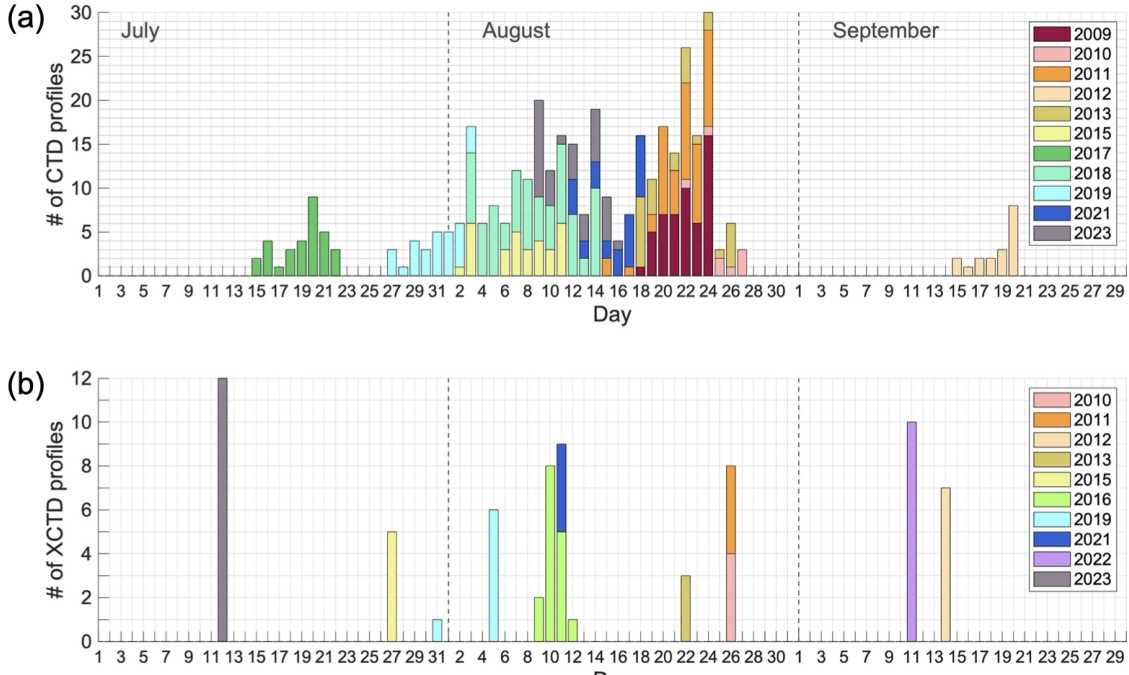

**Figure 3.** A visual representation of the survey dates and time periods in the summer season (July–September). Number of CTD (top) and XCTD (bottom) profiles collected on each day in the summer survey period across all years. Bar colors represent individual years. Dashed grey vertical lines denote the start/end of a month.

Prior to objective mapping, we perform a "bottom fill" procedure for profiles extending beyond 550 m or deeper to enhance
data density of the deep fjord regions for the objective mapping process. For all profile locations, properties below the sill depth
of 550 m show little variability and are remarkably stable with respect to depth, but they do vary in the along–fjord distance.
Without bottom filling of these profiles, the deepest profile informs the properties at that depth across the fjord when it is
more likely that properties are similar to their nearest vertical neighbors. First, for all profiles extending 550 m or deeper, we
calculated the average temperature and salinity value of the deepest 10 m of that profile. Then we extrapolated these properties
uniformly to the bottom. This extrapolation procedure was used for 128 profiles out of a total of 172 used in the along–fjord
sections.

## 4.2   Creating gridded data using objective mapping

The challenge of creating gridded fields from scattered observations is well known in the earth sciences and there are many
possible approaches. Objective mapping (also referred to as optimal or optimum interpolation) allows for the explicit use





of input parameters and use of multiple spatial correlation scales to better represent physical processes. Objective mapping approaches are commonly applied to other hydrographic profile datasets including from the northern Antarctic Peninsula (Dotto et al., 2021) and the Weddell Sea (Reeve et al., 2016), and of biogeochemical profiles in the Southern Ocean (Mazloff et al., 2023). These previous applications are concerned with larger ocean basin–scale observations, often involving thousands of profiles, and spanning decades. This is the first application of objective mapping for a Greenland fjord. Only recently have

we been monitoring Sermilik Fjord long enough (> 10 years) and with dense enough observations to appropriately inform the parameters and assumptions of the interpolation method. The increased utility provided by a gridded dataset became apparent as research about Greenland fjords is maturing and data volume is increasing.

The along–fjord sections constructed with the final selected profiles for each year were mapped onto a 2 km (horizontal) x 5 m (vertical) grid with the objective mapping procedure. This process was performed independently for temperature and

salinity variables. The depth extent of the objectively mapped grid for each year was determined by the depth of the deepest profile and the horizontal extent was determined by the minimum and maximum along–fjord distances of the profile locations for each location. This results in different sized grids bounded by the maximum observational extent for each year.

### 4.2.1 Objective mapping algorithm

First, a background field across the full domain (also referred to as a "first guess") was created by considering all profiles col-

lected in the fjord across all years. The resulting background field, $g_b$, represents the large–scale field which is well determined by the data and horizontal sampling locations.

For each year, the data anomaly $d'$ was calculated as

$$d' = d - Hg_b \tag{1}$$

where $d$ is a vector of the original profile data for one year with n number of total data points at unique locations $(x_i, z_i)$. $H$ is a matrix operator that linearly interpolates $g_b$ to the same spatial coordinates as the original data points of $d$.

The data anomalies for each year, $d'$, are then objectively mapped using the gain matrix, $K$, to produce the final gridded field by adding the background field, $g_b$, back to produce the final gridded field, $g_a$ (Eq. 2). It is common practice to objectively map the anomaly field (Bretherton et al., 1976; Roemmich, 1983). Prior to being mapped using Eq. (2), the anomaly data, $d'$,

were normalized by the standard deviation and the mean was subtracted.

The gridded field for each year, $g_a$, is produced by

$$g_a = Kd' - g_b. \tag{2}$$

The gain matrix, $K$, objectively maps the data anomalies, $d'$, and the background field is added back in. The core part of

the objective mapping procedure is $K$ (Ide et al., 1997). It's also referred to as the coefficient or weighting matrix (Wong et al., 2003; Reeve et al., 2016). $K$ is constructed using the data–data spatial covariance matrix, $C_{dd}$, and the data–grid spatial





covariance matrix, $\mathbf{C_{dg}}$, where

$$\mathbf{K} = \mathbf{C_{dg}} \cdot [\mathbf{C_{dd}} + \mathbf{R}] - 1. \tag{3}$$

We assume the covariances are each a sum of a large–scale Gaussian and a small–scale Gaussian (Eq. 4 and Eq. 4). The decay scales of each Gaussian are determined by four scale parameters: large– and small–scale horizontal correlation scales, $L_{x_1}$ and $L_{x_2}$; and large– and small–scale vertical correlation scales, $L_{z_1}$ and $L_{z_2}$. Each Gaussian has an amplitude parameter which determines the relative weighting of the large–scale ($A_1$) and small–scale ($A_2$) functions in the final map. The sum of $A_1$ and $A_2$ must be equal to 1 and must both be positive values. These six parameters were prescribed and determined by prior knowledge of scales of variability in the system, and tuned such that they yield a realistic field that best captures the conditions and dynamics of the fjord.

$$\mathbf{C_{dd_{ij}}} = A_1 \cdot \exp\left\{-\left[\frac{dist^2_{x_{ij}}}{L_{x1}} + \frac{dist^2_{z_{ij}}}{L_{z1}}\right]\right\} + A_2 \cdot \exp\left\{-\left[\frac{dist^2_{x_{ij}}}{L_{x2}} + \frac{dist^2_{z_{ij}}}{L_{z2}}\right]\right\} \tag{4}$$

$$\mathbf{C_{dg_{ig}}} = A_1 \cdot \exp\left\{-\left[\frac{dist^2_{x_{ig}}}{L_{x1}} + \frac{dist^2_{z_{ig}}}{L_{z1}}\right]\right\} + A_2 \cdot \exp\left\{-\left[\frac{dist^2_{x_{ig}}}{L_{x2}} + \frac{dist^2_{z_{ig}}}{L_{z2}}\right]\right\} \tag{5}$$

The data–data spatial covariance matrix, $\mathbf{C_{dd}}$, is a function of the distances between every i data point location to every other j data point location in the horizontal ($dist_{x_{ij}}$) and depth ($dist_{z_{ij}}$) directions. $\mathbf{C_{dd}}$ is thus a n x n square matrix where n is the number of original data points. The diagonal values of this matrix are 1 as this represents the distance of each profile data point to itself. The data–grid spatial covariance matrix, $\mathbf{C_{dg}}$, is a function of the distances between every i data point location to every g grid point location in the final gridded domain. $\mathbf{C_{dg}}$ is an m x n matrix where m is the total number of grid point locations.

To represent noise in the system, we define the noise matrix, $\mathbf{R}$, as

$$\mathbf{R} = \epsilon^2 \cdot \mathbf{I}. \tag{6}$$

The value of $\epsilon^2$ is a prescribed noise-to-signal parameter of the data anomalies. A larger value of $\epsilon$ means that the map is less able to represent the data anomalies and the final map is less influenced by those data values because more "noise" is assumed to exist in the system. The choice of $\epsilon$ strongly impacts the final map (Mazloff et al., 2023). Our choice here is $\epsilon = 0.5$ times the standard deviation of the data anomalies, $d'$, and this is a parameter that can be adjusted depending on application or the question of interest. $\epsilon = 0$ would represent that the values of the final map at the data locations must be equal to the original data values. However, it is not possible to use $\epsilon = 0$ because the sum of $\mathbf{C_{dd}} + \mathbf{R}$ in Eq. 3 produces a non-zero diagonal for the matrix inversion, which is mathematically necessary for this objective mapping procedure. A non-zero value for $\epsilon$ acknowledges that we are modeling the system with smoothness at the lengthscales specified in Eq. 4 and Eq. 5 and signals at smaller





scales than these are considered "noise" not represented by the chosen lengthscale parameters.


Objective mapping allows for the calculation of the error variance of the gridded field as

$$\boldsymbol{\sigma_g}^2 = \mathrm{diag}(\mathbf{I} - \boldsymbol{C_{dg}} \cdot [\boldsymbol{C_{dd}} + \boldsymbol{R}]^{-1} \cdot \boldsymbol{C_{dg}^T}) \cdot \boldsymbol{\sigma_d}^2. \tag{7}$$

$\boldsymbol{\sigma_d}^2$ is the variance of the original data, $\boldsymbol{d}$. We report uncertainty in units of the observed data by taking the square root of Eq. 7. We share the gridded fields of the uncertainty (also referred to as mapping relative error) so that a user can gauge the

amount of uncertainty in the map.

After deriving the gridded fields, we apply a correction to the temperature field. For the coldest temperatures at the freezing point of seawater with limited data points nearby, the objective mapping procedure interpolates the gridded temperature to be colder than the freezing point. We identify these values and correct them to be at the freezing point of seawater, calculated using the gridded salinity value of that gridded data point. This correction was applied to 27 out of 77,521 total gridded data

points in all the final gridded temperature fields.

### 4.3   Choosing a background field

The background field, $\boldsymbol{g_b}$, is an important parameter in the objective mapping method and can be constructed in a number of ways. The constant hydrographic features we aimed to capture in a background field were the two–layer temperature structure of the fjord (cold water on top of warmer water), colder surface waters in the ice mélange relative to the mouth, and a large

salinity range with nearly fresh water at the surface and increased salinity with depth. For both temperature and salinity fields, we plotted all selected along–fjord profiles from every year in the fjord onto one along–fjord axis, linearly interpolated these profiles to the 2 km x 5 m standardized grid. We then heavily smoothed this grid in the vertical and horizontal using a boxcar filter to achieve the desired largescale background field features. We explored alternative methods for choice of background field, including creating individual background fields for every year from linearly interpolating between spatial endmember

profiles closest to the glacier and mouth. We determined that any reasonable method for creating a background field that results in a representation of the average large–scale hydrographic features described above leads to similar objective mapping performance.

### 4.4   Choosing appropriate parameters

In the objective mapping method used here, there are seven parameters chosen based on knowledge of the system and the

observations (Table 3). The same parameters were used for every year and for both temperature and salinity fields. Other applications of objective mapping in fjords could use more or fewer Gaussian lengthscale functions with different values and relative weights in order to capture the dynamics of a specific system. Initial lengthscale values were tested based on visual inspection of the scattered data and an error value was chosen informed by previous objective mapping applications and known instrument error (Dotto et al., 2021; Mazloff et al., 2023). Following other studies, we performed a series of tests exploring the

parameter space to choose a parameter combination that yielded minimized residual values and an appropriate representation




of smoothed hydrographic features for the whole domain each year. By making the code for this method available, we stress that others can adjust parameters to produce gridded fields that best match features of interest within Sermilik Fjord or best match features in other fjords with different spatial patterns.

**Table 3.** Description of assigned parameters in objective mapping method

| $L_{x_1}$ (km) | $L_{z_1}$ (m) | $A_1$ | $L_{x_2}$ (km) | $L_{z_2}$ (m) | $A_2$ | $\epsilon$ |
|---|---|---|---|---|---|---|
| 50 | 100 | 0.6 | 15 | 10 | 0.4 | 0.5 |

## 4.5 Calculating a climatology

We calculated a climatology representing the average summer state properties of Sermilik Fjord from all the yearly gridded data. For both temperature and salinity, the mean and root mean square deviation (RMSD) at every grid cell was calculated. The 2009–2023 yearly grids cover different fjord extents based on the locations of the original profiles for each year. As a result, each grid cell mean and RMSD was calculated from a different number of years ranging from 1–14 (Fig. 4). The 2023 CTD and XCTD data are not combined and treated as two separate instances of a summer fjord state, yielding a maximum of 355 14 available grids to be used in calculating each grid cell mean. Grid cells between 40–50 km from the terminus use all 14 grids while grid cells 12 km from the terminus only have 3 grids to calculate the means. This reflects the challenge of obtaining repeat observations in the ice mélange region. Grid cells with fewer than 3 years were not included in the final climatology.

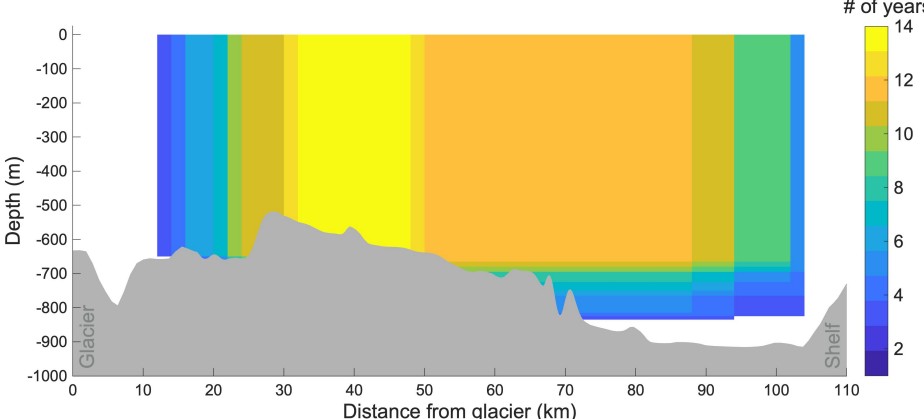

**Figure 4.** The number of yearly grids that cover each grid cell in the along–fjord standardized grid domain. The number in each grid cell is used to calculate the property mean and RMSD. Grid cell size is 2 km (horizontal) by 5 m (vertical). The 2023 CTD and XCTD data are not combined and instead treated as two separate instances of a summer fjord state leading to a total of 14 possible grids. Grid cells with only 1 or 2 years of coverage have been removed.



## 5 Results

### 5.1 Objective mapping performance

We judged the parameter choice performance based on the root mean squared error (RMSE) values, the pattern of residuals across the domain, and the visual representation of hydrographic features in the final gridded fields. Residuals were calculated by subtracting the gridded value from the observed value, where the gridded data were linearly interpolated to match the exact along–fjord and depth coordinates of each observed data point. The gridded temperature fields at the profile locations, residuals at those locations, and the associated temperature relative mapping error for the years 2009 and 2019 are shown in

Fig. 5. We highlight these two years with differing hydrographic patterns and spacing between profile locations to demonstrate the objective mapping performance across a range of input profile data characteristics.

We assessed the residuals and RMSE for each year individually and across all years in bulk for both temperature and salinity fields. For the final parameter set presented here for all years, 88of temperature residuals are within a difference range of $\pm$ 0.1 °C and 67are within a difference range of $\pm$0.04 °C. 91of salinity residuals are within a difference range of $\pm$ 0.05 g kg$^{-1}$

and 77are within a range of $\pm$ 0.02 g kg$^{-1}$ (Fig. 6). The bulk RMSE is 0.09 °C and 0.08 g kg$^{-1}$, for temperature and salinity respectively. The observed salinity profiles are more spatially smooth compared to the temperature profiles resulting in slightly smaller RMSE values.

The pattern of residuals across the domain is equally important to understand which hydrographic features are and are not represented well in the interpolation method. Sharp thermoclines within the upper 200 m at length scales smaller than the

prescribed vertical smoothing ($L_{x_2}$ = 10 m) result in the highest residual values in the domain. These are often present near the fjord mouth where interleaving of different water masses between the fjord and shelf is known to occur. High residuals also consistently occur in the ice melange region at 150 m depth where there is a sharp thermocline transition between cold ice mélange meltwater and warmer waters at depth.

We also provide the mapping relative errors in the same format as the temperature and salinity gridded fields. The user is able

to choose a relative error value to work with. Considering all years the range of error is 0.03 to 0.30 °C and 0.04 to 0.40 g kg$^{-1}$ for the temperature and salinity grids respectively. As expected, the error increases as distance between data points increases beyond the lengthscale parameters.

Finally, parameter choice was aided by visual inspection of the original profiles and gridded data for each year in conservative temperature–absolute salinity ($\Theta$–$S_A$) diagrams (Fig. 7). This was to ensure that the gridded data did not introduce density

classes beyond the range present in the observations. The final parameter set shows good agreement between the gridded data and observations in $\Theta$–$S_A$ space.

### 5.2 Climatology and spatial variability

The complete set of conservative temperature ($\Theta$) and absolute salinity ($S_A$) along–fjord gridded sections are available to view in the Supplementary Material. Grids from 2009 and 2019 are shown here as examples of the gridded data (Figs. 5, 7).

Combining the CTD and XCTD datasets extends the along–fjord spatial coverage for years where both types of profiles were

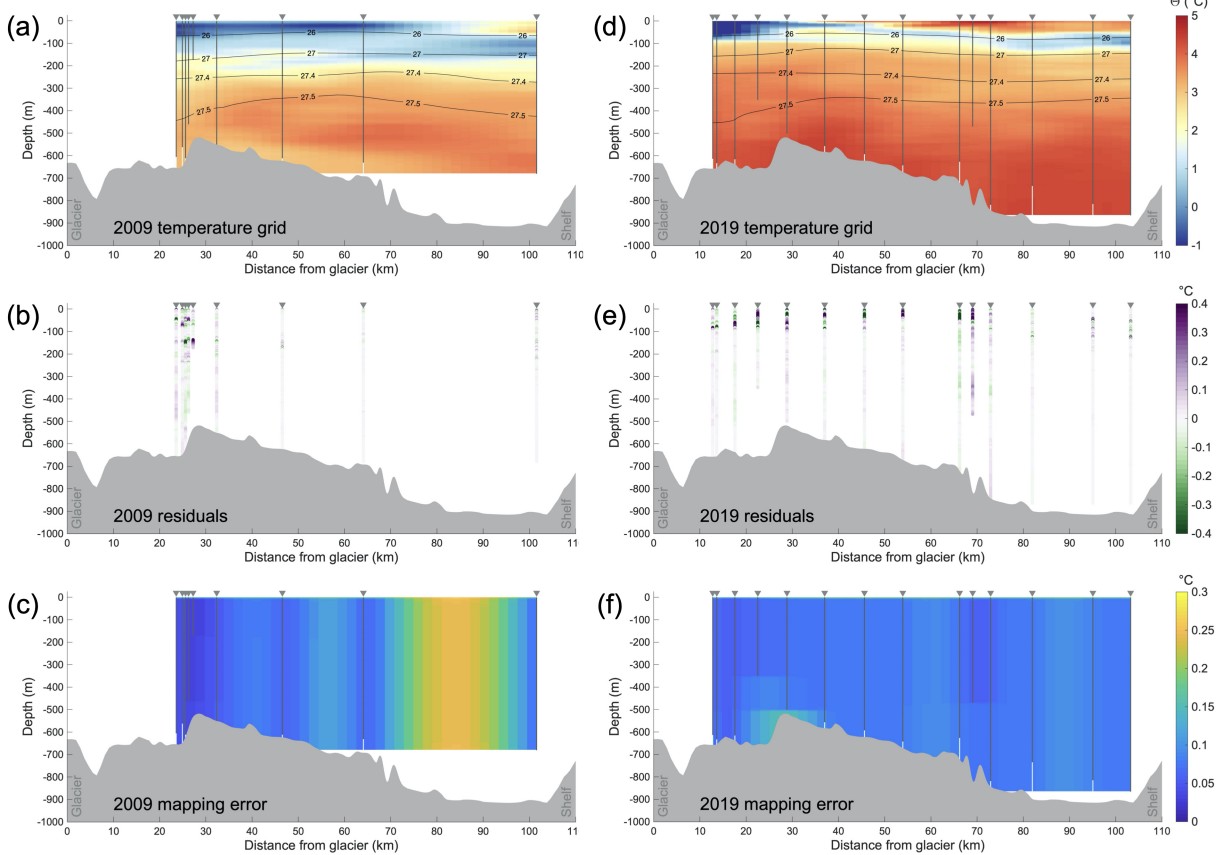

**Figure 5.** The conservative temperature objectively mapped gridded fields for 2009 (a) and 2019 (d) with potential density anomaly isopycnals (black contours). Original CTD and XCTD profile locations in the along–fjord direction are shown by triangle markers with vertical black lines representing the depth extent of each profile. White lines show where the bottom of a profile was filled in to facilitate greater data coverage for mapping. The 2009 (b) and 2019 (e) residual values are shown by filled dots at each original data point. The mapping relative error (c, f) is shown for each year. The bathymetry (grey area) is derived from a thalweg line (Figure 1) of BedMachine Greenland v4 data (Morlighem et al., 2017). Distance 0 km is the location of the Helheim Gletsjer terminus in 2019.

collected concurrently. Despite differences in original profile locations and fjord coverage from year to year, the standardized gridded along–fjord sections allow us to calculate a summer state climatology and associated RMSD for the Helheim–Sermilik Fjord system (Fig. **??**). This is one example of the increased utility of the gridded dataset compared to utilizing individual surveys or less explicit interpolation methods. The climatology product provides novel context for the yearly variability of
summer season fjord water properties and improves the interpretation of previously published work from Sermilik Fjord.



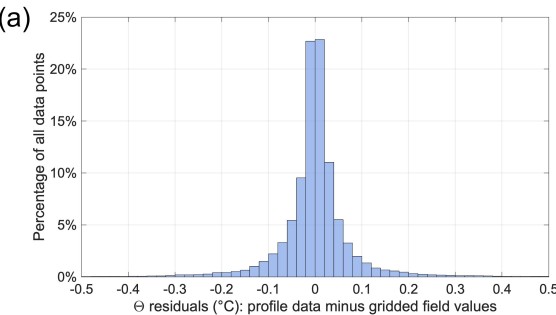 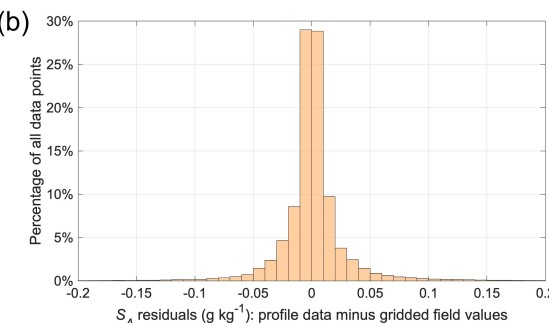

**Figure 6.** Histograms displaying the distribution of residual values between the gridded temperature (a) and salinity (b) fields and the profile observations considering all years in bulk. Bin size is 0.02 °C (a) and and 0.01 g kg⁻¹ (b)

Typical of Greenland glacial fjords, the climatology shows salinity is the dominant driver of stratification. The basic salinity structure is consistent throughout the fjord, with a maximum salinity of 34.95 g kg⁻¹ below 400 m, gradually freshening toward 34.40 g kg⁻¹ at 200 m depth (Fig. 8b). The halocline steepens in the upper 200 m, with the surface layer having the sharpest vertical salinity gradients over the entire along–fjord extent.

Below 400 m, temperature is also relatively uniform along the length of the fjord (Fig. 8a). The mean properties of this layer (3.81 ± 0.15 ° C, 34.89 ± 0.04 g kg⁻¹ , potential density anomaly $\sigma_0$ > 27.5 kg m⁻³) are consistent with established characteristics of inflowing AW from the continental shelf (Straneo et al., 2011; Jackson and Straneo, 2016; Beaird et al., 2018; Lindeman et al., 2024).

    Above 400 m, we see more spatial variability in temperature indicative of different water masses and ice–ocean processes.

To identify characteristics of along–fjord variability above 400 m, we have separated the fjord into three regions based on the water mass properties and established process understanding. We proceed by first describing water properties at the mouth region where we expect the fjord to be influenced by exchanges with the shelf. Second, we describe the near glacier region and address the glacial forcing on water properties, and finally we describe the mid–fjord region which shows gradients between the shelf and near glacier water properties.

### 5.2.1   Fjord mouth properties

At the fjord mouth (averaged over 94 km–104 km from the glacier), we see a mean temperature structure similar to the established typical summer properties on the continental shelf nearby. Below a near–surface warm layer, there is a subsurface temperature minimum (50 m–100 m, $\sigma_0$ = 26–27 kg m⁻³), which is a similar depth and density range to the cold PW layer typically observed on the shelf. However, the mean temperature in this layer (0.5 °C–1 °C) is warmer than typical PW on the

shelf (<0 °C) (Sutherland and Pickart, 2008). Separate from the gridded data, we used the observed profiles from the shelf region to calculate the mean PW properties on the shelf. For each year, we created a singular representative shelf profile by taking the temperature minimum of each isopycnal band across all shelf profiles. We then calculated a mean profile from the

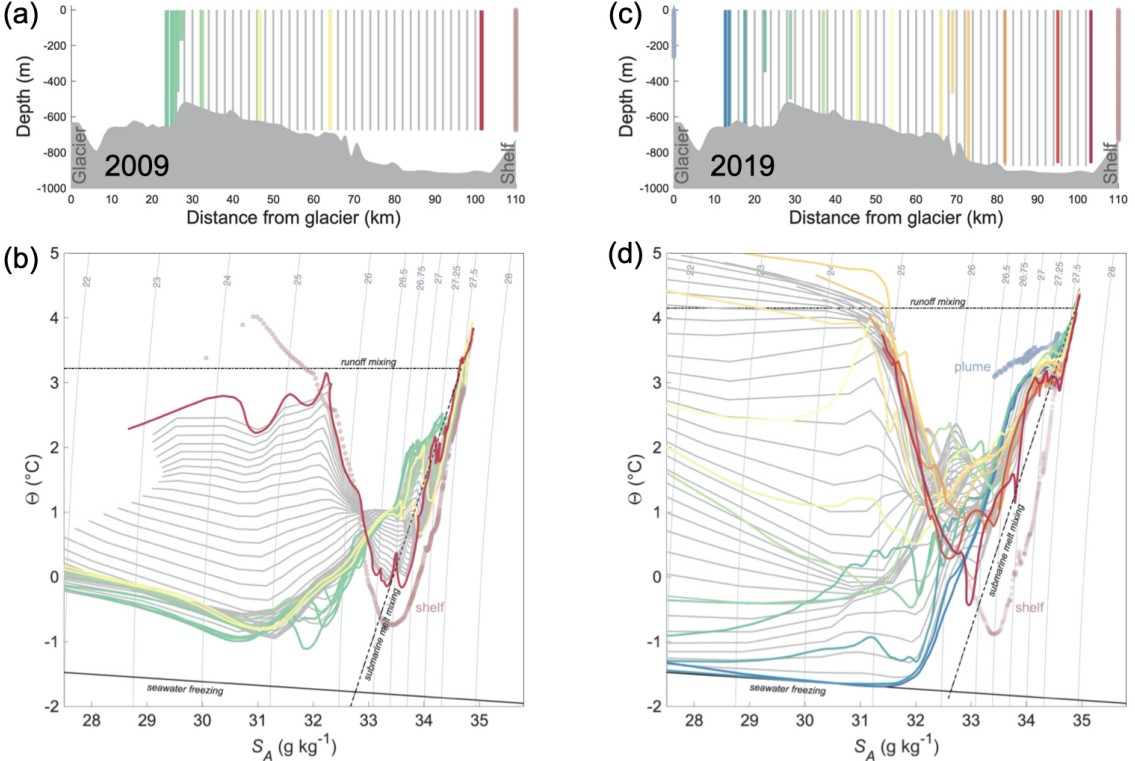

**Figure 7.** Comparisons of original profiles and gridded data for years 2009 (a, b) and 2019 (c, d) in $\Theta-S_A$ diagrams (b, d). Panels (a) and (c) show the along–fjord location of the original profiles, colored by distance from the glacier, in comparison to location of the grid cells (grey lines). $\Theta-S_A$ diagrams (b,d) show original profiles, colored by distance from the glacier in panels (a) and (c). Grey lines are the gridded data with every column in the grid plotted as an individual profile. Shelf profiles from both years are shown in light pink and two plume profiles from 2019 are shown in light blue. Mixing lines are plotted for submarine ice melt (dashed line) and melt runoff (dot–dash line) and the seawater freezing line is the black solid line. Grey contours are potential density anomaly isopycnals in kg m$^{-3}$.

representative shelf profiles of each year (Fig. 9). The subsurface temperature minimum of this profile indicates the core of the PW layer present on the shelf. The mean shelf PW properties for all years are -1.22 $\pm$ 0.35 °C and 33.17 $\pm$ 0.25 g kg$^{-1}$

occurring at a mean depth of 77 m and ranging from 50 m– 115 m depth.

At the mouth, a thermocline centered on the 27 kg m$^{-3}$ isopycnal separates the subsurface temperature minimum from the underlying warm AW. The AW properties at the mouth vertically averaged between 400m–700 m are 3.85 °C $\pm$ 0.13 °C and 34.88 g kg $\pm$ 0.04 g kg$^{-1}$.

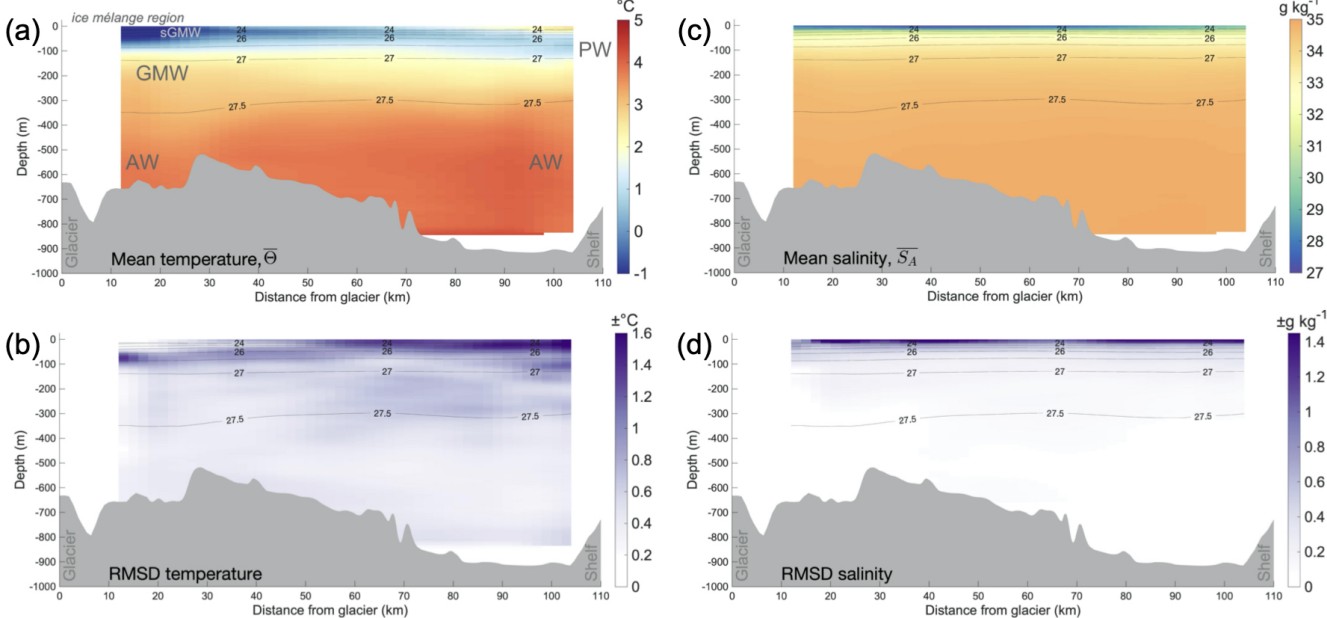

**Figure 8.** Gridded 13 year mean summer state climatology of conservative temperature (a) and absolute salinity (b) of the for Sermilik Fjord derived from hydrographic observations between 2009 to 2023. Associated RMSD about the mean for conservative temperature (c) and absolute salinity (d) show an estimate of spread of values in time at every grid cell. The $\sigma_0$ = 24, 26, 27 and 27.5 kg m$^{-3}$ potential density anomaly isopycnals are represented by grey contours in every panel and water masses discussed in the text (Atlantic Water (AW), glacially modified water (GMW), Polar Water (PW), and surface GMW (sGMW)) are labeled in panel (a).

### 5.2.2 Near glacier properties

In the near glacier ice mélange region (horizontally averaged between 12 km–22 km from the glacier), the climatological mean AW properties vertically averaged between 400 m–700 m are 3.59 °C $\pm$ 0.17 °C and 34.84 $\pm$ 0.05 g kg$^{-1}$, within one standard deviation of the properties at the mouth.

Above the AW layer, the near glacier region has a positive temperature anomaly when compared to the mouth (between 75 m–300 m) and shelf (between 55 m - 300 m), calculated by subtracting along isopycnals. The temperature anomaly and

features of the near glacier profiles in $\Theta$–$S_A$ space (Fig. 9) are characteristic of GMW (Straneo et al., 2011; Muilwijk et al., 2022). Building on previous work, we can identify the fingerprints of different freshwater sources and determine their relative importance for setting the GMW water properties throughout the fjord.

We use the mixing lines displayed on the $\Theta$–$S_A$ plot to differentiate between SGD, surface runoff, or SMW influence (Fig. 9). The runoff mixing line shown represents the expected water properties for the mixing of averaged unmodified AW at 550

m and SGD (with assumed properties of 0 °C and 0 g kg$^{-1}$). The SMW mixing line, or Gade line, represents the combined influence of latent heat uptake and mixing with meltwater (Gade, 1979). The two XCTD profiles collected directly in the plume

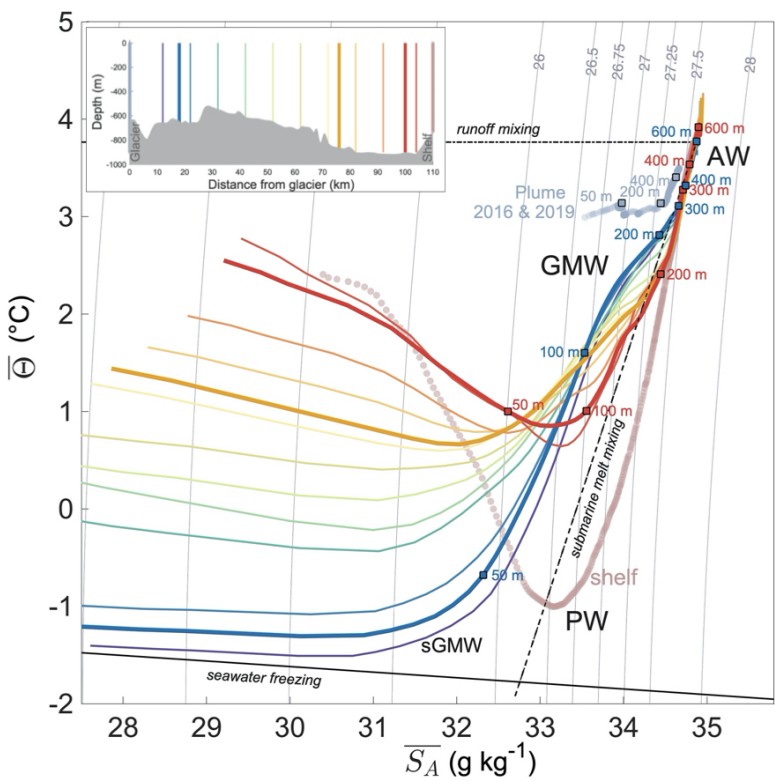

**Figure 9.** Conservative Temperature–Absolute Salinity ($\Theta$–$S_A$) diagram of the gridded summer state climatology for Sermilik Fjord. Grid cells every 10 km in the along–fjord direction are plotted, with cool to warm colors representing increasing distance from the glacier terminus. Bold lines are representative of the three regions (mouth, mid–fjord, and near glacier) discussed in text and referred to in Figs. **??**. The mean representative shelf profile is plotted in faded pink dots. The average of profiles from the subglacial discharge plume polynya in 2016 and 2019 are faded blue dots. Mixing lines are plotted for submarine ice melt (dashed line) and melt runoff (dot–dash line) and the seawater freezing line is the black solid line. Grey contours are potential density anomaly isopycnals. Water masses discussed in text (Atlantic Water (AW), glacially modified water (GMW), Polar Water (PW), and surface GMW (sGMW)) are labeled.

at the glacier terminus from 2016 and 2019 strongly parallel the runoff mixing line slope. Between 165m–300m (27.22 kg m$^{-3}$ < $\sigma_0$ < 27.49 kg m$^{-3}$), the slope of the near glacier profiles (20 km from the glacier terminus; dark blue profiles in Fig. 9) deviate in the composite direction of the runoff mixing line and submarine melt line. This is consistent with a mixture of plume waters and SMW from the ice melange. The inflection of the slope at depths shallower than 165 m ($\sigma_0$ < 27 kg m$^{-3}$) suggests that this is the upper limit of SGD, which can be used as an estimate of the climatological average neutral buoyancy depth range of plume waters in this region of the fjord.

Between 50 m and 165 m (26.0 kg m$^{-3}$ < $\sigma_0$ < 27.22 kg m$^{-3}$), the slopes of the near glacier profiles directly parallel the submarine melt mixing line, indicating the addition of SMW from the ice melange as a primary driver of water properties




at these depths. Above 50 m, the $\Theta$–$S_A$ properties converge toward the local freezing temperature as the near surface waters are both cooled by SMW and freshened by surface runoff. The surface water properties nearest to the glacier (12 km) are the coldest in the entire domain, with a minimum temperature of -1.49 °C at 15 m depth. This cold pool is characteristic of the ice mélange region, which extends 30 km into the fjord. This also leads to the near glacier region showing the strongest surface stratification of all the regions (Fig. 10). Following Lindeman et al. (2024), we identify this as sGMW occurring at less than

50 m depth where $\sigma_0 < 26$ kg m$^{-3}$.

### 5.2.3 Mid–fjord properties

Considering the characteristics of the mid–fjord region (40 km–90 km from the glacier) in $\Theta$–$S_A$ space reveals that the water properties are a progressive mixture between the two endmembers of the mouth and near glacier profiles (Fig. 9). We assume along–isopycnal mixing is occurring in the upper 400 m of the mid–fjord region ($\sigma_0 < 27.5$ g kg$^{-1}$) where GMW is being

exported down the fjord and meeting waters of the same density coming into the fjord from the shelf. Above 50 m, the surface waters are warmer than the sGMW found in the melange, with minimum temperatures above 0 ºC.

The AW properties in the mid–fjord are similar to those at the mouth. The mid–fjord AW properties vertically averaged between 400 m–700 m and horizontally averaged between 60 km–70 km are 3.83°C $\pm$ 0.05 °C and 34.90 g kg$^{-1}$ $\pm$ 0.02 g kg$^{-1}$.

The middle of the fjord is at 60 km in the along–fjord distance coordinate axis. However, the map view of the fjord geometry

is defined by a constriction at 75 km (at 65.60 ºN in Fig. 1) which influences the exchange of waters between the fjord mouth and upper fjord. Previous studies have described mooring records at this location (Jackson et al., 2014; Jackson and Straneo, 2016; Snow et al., 2023). To easily compare the findings of this study with the moored data, we define "mid–fjord" as 76 km. As Figs. **??** show, the water properties are similar between 60 km and 76 km from the glacier.

### 5.3 Variability of yearly gridded data

While the climatology shows the average summer hydrography, there is yearly variability and unique hydrographic patterns represented in the individual yearly grids which are important to consider (see Supplement to view all yearly grids). The temporal RMSD of the properties climatology grids (Fig. 8c, d) provide an initial sense of the year–to–year variability in different regions of the fjord. Below 400 m, in the largely unmodified AW layer for the whole along–fjord domain, we see consistent RMSD values. The spatial average of the RMSD between 400 m–700 m and horizontally from 12 km–104 km are

0.39 $\pm$°C and 0.06 g kg$^{-1}$.

The year–to–year variability above 400 m is relatively greater, with maximum RMSD values of both temperature and salinity occurring at the surface. While salinity RMSD vertical structure is similar between the fjord mouth to the near glacier region, the temperature RMSD vertical structure varies along the fjord above 400 m and most prominently in the surface layers.

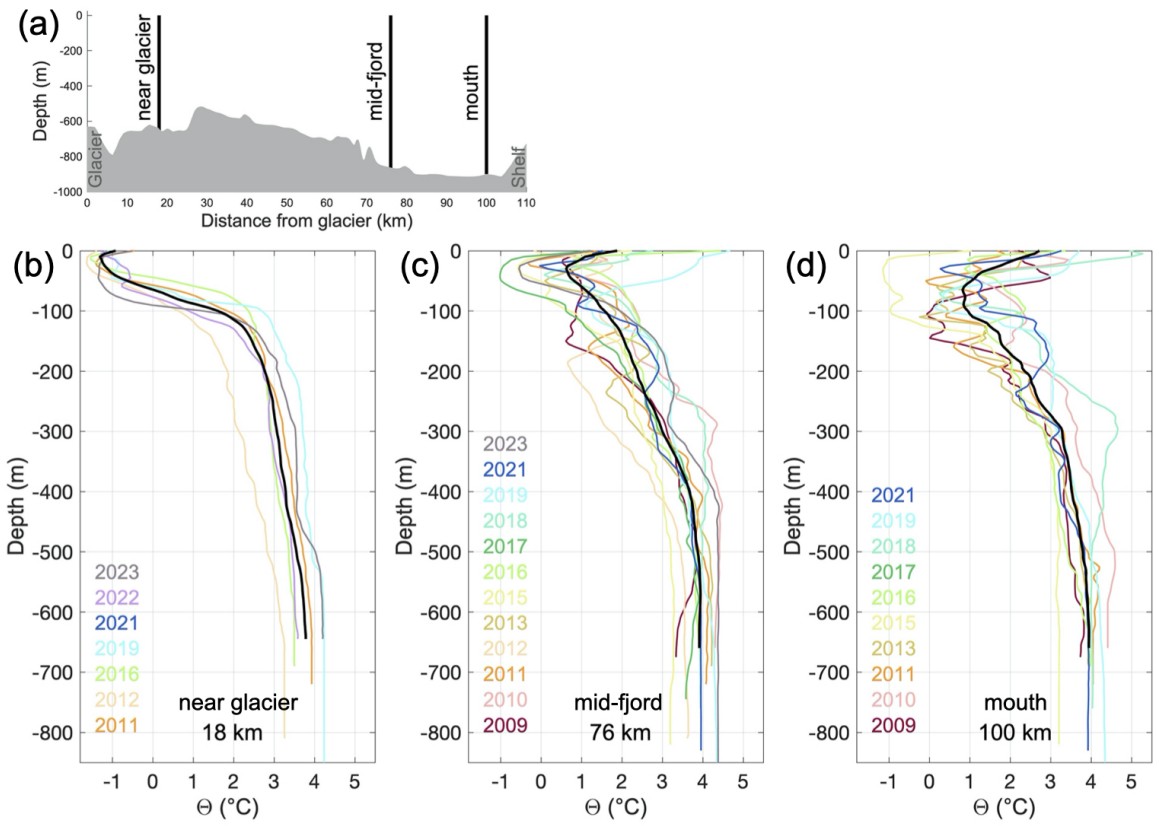

**Figure 10.** Conservative temperature profiles from all the yearly grids at locations shown in (a): the near glacier (b), mid–fjord (c), and mouth regions (d). Colored lines represent individual years and the bold black line is the time mean climatology profile.

### 5.3.1 Fjord mouth variability

The mouth region shows the most variability from year to year (Fig. 8c). We speculate this is due to shelf processes, occurring on multiple timescales from seasonal to daily, influencing the properties measured at the mouth during a given field campaign (Jackson and Straneo, 2016; Jackson et al., 2018). The coldest water of the fjord mouth region in 2015 and 2017 matches the PW shelf properties and depth range for each year. In other years (2010, 2018, 2019, 2020), shelf PW properties are not present in the fjord. Many of the mouth profiles show characteristic interleaving patterns above 200 m, where shelf and fjord waters

of similar density are meeting and mixing. From $\Theta$–$S_A$ plots of individual years, we see evidence of GMW mixing with shelf PW at the mouth region for some years (Supplemental Figures). Note that the average mouth profile is less smooth compared to the average near glacier profile due to multiple intrusions with sharp thermoclines at various depths being averaged together. This also impacts the stratification, which shows the largest range of yearly values between 50–200 m at the mouth (Fig. 11).



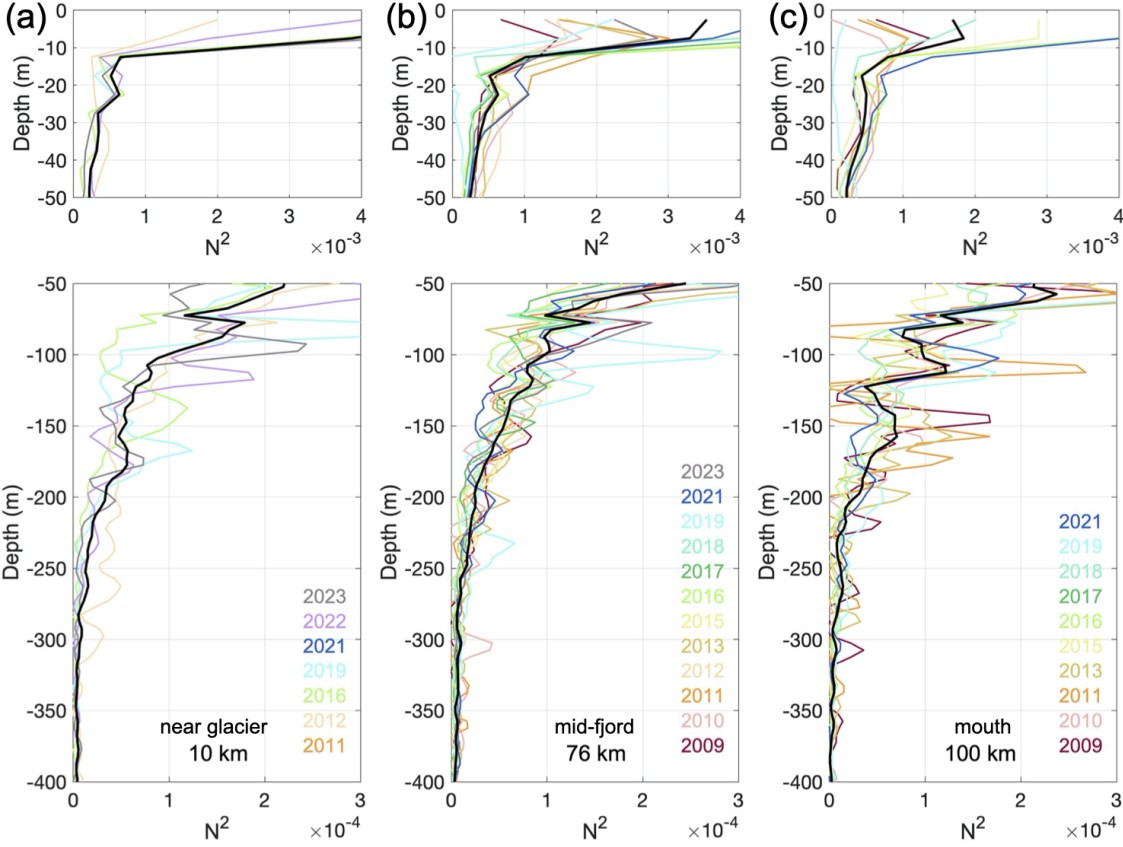

**Figure 11.** $N^2$ (Brunt–Vaisala frequency), a measure of stratification, from the near glacier (a), mid–fjord (b), and mouth (c) regions for all yearly gridded data. The profiles are from the same locations as in Figure 10. The upper panels have a different vertical scale to emphasize the surface. The horizontal scale in the upper panels is an order of magnitude larger the lower panels. $N^2$ was calculated from conservative temperature and absolute salinity gridded data using the TEOS–10 Oceanographic Toolbox (McDougall and Barker, 2011). Black bold lines represent the mean $N^2$ profile for all years in each region.

### 5.3.2   Near glacier variability

In the near glacier region, the upper 50 m has the smallest temperature RMSD of the fjord domain. This highlights the consistency of the ice mélange in setting the temperature properties in this area. From 50 m–100 m, there is an increase in the temperature RMSD (maximum $\pm$ 1.09 °C) as the thermocline is more variable– likely in response to variability in GMW extent in the water column and properties. Below the thermocline, the average RMSD of the GMW layer properties between 165 m–300 m are $\pm$ 0.51 °C and $\pm$ 0.14 g kg$^{-1}$. As previous studies have noted, the properties of this GMW layer vary with the
unmodified AW properties at depth near the glacier as these are the waters being upwelled (Muilwijk et al., 2022). When AW





near the glacier is cooler (warmer) than average for a given year, the GMW properties resulting from upwelling in the SGD plume are cooler (warmer) than average.

### 5.3.3 Mid–fjord variability

Profiles from the mid–fjord region for each year show similarly variable interleaving and intrusion features as the mouth profiles, though the mid–fjord intrusions are less sharp overall. Some years, properties in the mid–fjord region are more similar to an average near glacier profile (2015, 2017, 2023) with < 0°C temperatures in the upper 50 m and exhibiting a strong thermocline between 50 m–100 m and relatively weak intrusions. Other years are more similar to a mouth profile (2011, 2012, 2021) with warmer surface waters and multiple stronger intrusions. The along–fjord location of the transition between near glacier properties and mouth properties varies from year to year. This variability is not captured in the time mean mid–fjord profile and climatology of the mid–fjord region.

### 5.3.4 Yearly anomalies from the mean

To further investigate the temporal variability, we subtracted the climatology from each yearly grid to produce the temperature and salinity anomaly for each year (selected years shown in Fig. 12 ). The temperature anomaly fields show years where nearly the entire fjord domain is warmer (e.g. 2019) or colder (e.g. 2015) than average. 2021 is the year with the smallest average anomaly across the whole domain, however, there are still areas in the domain that are warmer and colder than average.

The pattern in the gridded anomaly field can be both due to variability in the properties and variability in the depth of the thermocline. For example, the properties of the temperature minimum at the mouth could match the mean temperature minimum properties, but if the thermocline is at a different depth the anomaly will be nonzero.

## 6 Discussion

Hydrographic data from fjords is relevant to a range of users– from climate and earth scientists interested in ice–ocean dynamics to local communities that rely on the ecosystem. As such, there is a need to make fjord data available and useable by a wide range of users. Here, we show how repeat summer surveys of Sermilik Fjord can be used to construct gridded along–fjord sections from disjointed and discrete profile observations. These gridded data are more intuitive for understanding hydrographic patterns and facilitate comparisons of different surveys. We calculated a climatological mean, complemented by information about the range and patterns of variability, which improves our understanding of Sermilik Fjord and is useful for future dynamical studies. In the following sections, we identify several important considerations regarding using these gridded data and/or profile observations from the individual surveys.

### 6.1 Daily to interannual variability

While all the data are from the summer season (July–September), each yearly along–fjord section is a snapshot capturing the combined influence of processes occurring on a range of timescales (days, months, years). Single wind events within the

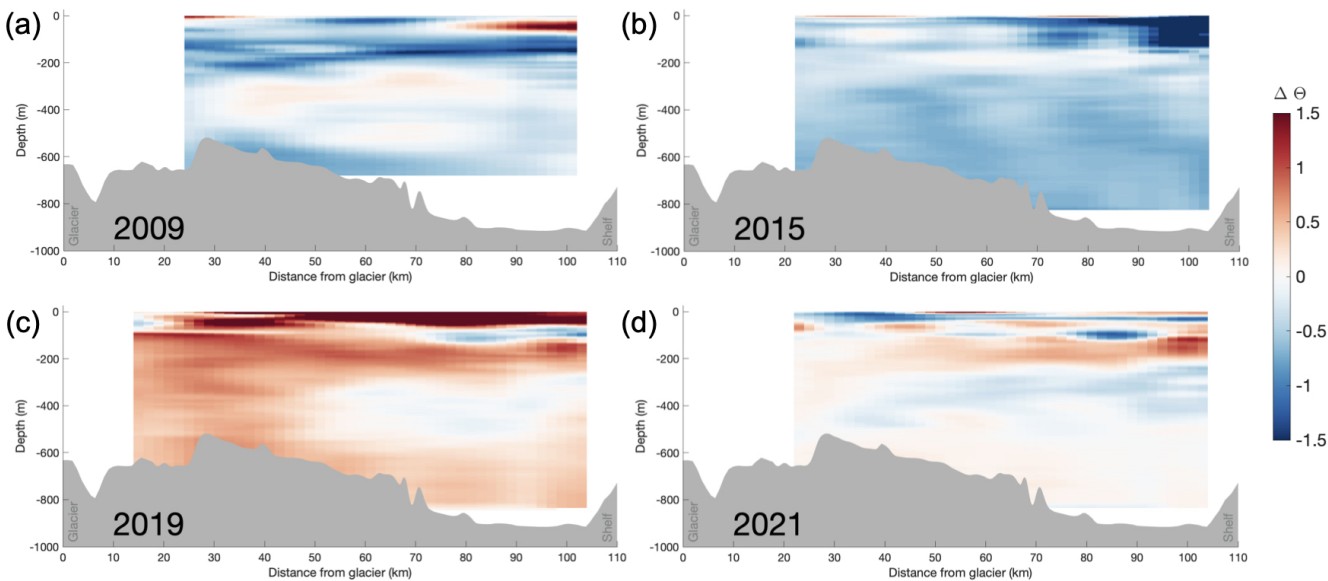

**Figure 12.** Conservative temperature anomalies from the climatological mean for yearly grids 2009 (a), 2015 (b), 2019 (c), and 2021 (d).
Cooler (warmer) colors show values less (greater) than the mean for each grid cell.

fjord and the adjacent shelf can influence fjord exchange and hydrographic properties over a timescale of days (Jackson and
Straneo, 2016). Integrating data collected before and after a single wind event during one survey can be challenging and creates
discontinuities in the domain. Some profiles were not used in the construction of a yearly along–fjord section because of these
discontinuities.

All data and yearly gridded products should also be interpreted within the context of the seasonal runoff cycle relative to
the timing of data collection. Fjord properties change dramatically over the course of the surface melt season, with subglacial
discharge input typically beginning in June and reaching its peak in August. Earlier studies have shown a progressive acceler-
ation of fjord circulation, and associated modification of fjord properties, as a result of this seasonal forcing (Sanchez et al.,
2023). For 2023, we did not combine the XCTD (collected on July 12) and CTD (collected August 9–16) surveys because

the fjord properties had evolved significantly in the time between surveys. These data are treated as two different along–fjord
sections. Because this data set incorporates sections collected at different points in the summer season (Figure 3), the temporal
variability reflects both interannual variability and the intraseasonal development of fjord conditions over the discharge period.

Variable regional atmospheric and ocean conditions occurring on larger spatial scales and longer time scales must also be
considered. For example, in 2015, there was significant sea ice within the fjord and on the shelf during the summer survey,

indicative of a prolonged winter. This is supported by findings that 2015 had an anonymously cold and long winter associated
with a positive state of the North Atlantic Oscillation in Southeast Greenland and the Irminger Sea (de Jong and de Steur, 2016).





Making any conclusions about interannual variability from this dataset must be done within the context of understanding the combined effect of these multiscale forcings on hydrographic patterns of each year. This will be the subject of future studies, and we are careful not to attribute hydrographic patterns discussed here to particular forcings as this requires further analysis
outside the scope of this work.

## 6.2   Water mass definitions and comparison to previous work

The average summer water mass properties calculated here are consistent with previously reported definitions of summer AW, PW, and GMW in Sermilik Fjord from CTD, XCTD, and moored observations (Straneo et al., 2011, Jackson and Straneo, 2016, Beaird et al., 2018, Sanchez et al., 2021, Lindeman et al., 2024). Notably, CTD data from 2009 (Straneo et al., 2011),
2015 (Beaird et al., 2018), and 2021 (Lindeman et al., 2024) was used in previous studies to identify the properties and depth range of GMW in these individual years. Reinterpreting the conclusions of these previous results within the context of the climatological mean and long term dataset is now possible. For example, Beaird et al. (2018) relied on the 2015 CTD survey. We now know that this is a year when nearly the entire fjord domain was 1°C colder than the 13 year climatological temperature mean (Figure 13b).
How water mass properties are defined and averaged has varied between studies based on particular applications. Different isopycnal ranges, depths, and/or horizontal extents are used to average and report water properties. The availability and format of the gridded sections allows a user to calculate any quantity they may need for a specific spatial extent or particular years based on the research question of interest. The gridded sections can also be combined with previously reported velocity data if a user wishes to calculate transport weighted means (Jackson and Straneo, 2016; Beaird et al., 2018).

**6.3   Gridded products facilitate model use and comparison**

Robust climatological means and consistently gridded surveys provide quantitative means for forcing or validating models. This is preferred over the use of single surveys or ad hoc choices made by earlier studies which used temperature and salinity profiles from Sermilik Fjord as initial conditions of idealized two–layer fjord models, boundary conditions and validation data for more complex numerical models, and ocean conditions for iceberg melt models and plume models (Sciascia et al., 2013;
Moon et al., 2018; Davison et al., 2022; Schild et al., 2021; Sanchez et al., 2024). The gridded products presented here now make it easier to find average conditions for different fjord regions depending on the research question, model initialization, and time period of interest.

While we have not included the plume polynya profiles in the gridded products, the individual profiles are available in the original data and these can be useful in studies employing plume models.

**7   Conclusions**

The dataset and gridded products presented in this study provide a crucial step toward standardizing and centralizing long–term fjord observations in Greenland. By compiling 13 years of hydrographic data from Sermilik Fjord, we offer a comprehensive





and accessible resource for studying fjord dynamics and ice–ocean interactions. The combined CTD and XCTD observations lead to greater spatial coverage of the fjord, including the melange region for multiple years and the subglacial discharge plume

polyna region for two years. The objective mapping method used to generate gridded fields is adaptable for different variables and fjord settings and can facilitate interdisciplinary disciplinary research– enabling comparisons with models, biological data, and other observations. Using the gridded fields, we shared a summer season climatology of Sermilik Fjord. This has provided necessary and new context for interpreting previous studies of Sermilik Fjord. We demonstrated how other quantities (eg. $N^2$) and water properties of specific regions can easily be calculated from the yearly gridded fields depending on questions of

interest.

This work highlights the need for a coordinated approach to fjord data collection and sharing. Establishing a structured, FAIR–compliant data repository for Greenland fjords will improve the accessibility and utility of these critical datasets, ultimately enhancing our understanding of glacial fjord systems and strengthening collaboration within the international science community and with Greenlandic partners .

## 580   8   Code and data availability

The gridded data products for each individual year and the climatology are available at the Arctic Data Center (https://doi.org/ 10.18739/A2513TZ0P, Roth et al. 2025) and GitHub (https://github.com/a1roth/sermilik_gridded_hydrography) as netCDF files. The files contain three dimensional gridded variables of conservative temperature and absolute salinity, as well as their respective mapping relative error matrices. Derived variables, like potential density or $N^2$, can be calculated by the user. Depth

levels (meters) and the thalweg along–fjord gridded coordinates in distance from the Helheim Gletsjer terminus position in 2019 and latitude and longitude are included in the file.

The original CTD and XCTD profiles from every year of sampling are all available at the Arctic Data Center as netCDF files (https://arcticdata.io/catalog/portals/sermilik/Data). Individual entries and DOIs have been created for each field campaign. All files include in situ temperature and practical salinity. Some years have additional variables from the CTD rosette, such as

dissolved oxygen and turbidity. All files include latitude and longitude coordinates of every profile and standardized depth levels (meters). As more hydrographic surveys are conducted in Sermilik Fjord, we plan for the data to be archived in this format and available at the Arctic Data Center in the Sermilik Hydrography Data Portal.

The code developed to create gridded along–fjord sections is available at GitHub (https://github.com/a1roth/sermilik_gridded_ hydrography). While this code is set up for Sermilik Fjord profiles, it can easily be adapted to other regions with discrete pro-

files in along–fjord sections. The parameters of the objective mapping method can be manually adjusted for different length scales and error input. Ancillary code for plotting and deriving other variables is also available.

*Author contributions.*   All authors designed and conceptualized the study. AR led the study, data processing, and writing of the manuscript. FS initiated and has led the data collection, processing, and research activities in Sermilik Fjord since 2008. FS, JH, and ML have all substantially



contributed to data processing, archiving, and manuscript writing. MM contributed code and guidance for the objective mapping method and
contributed to manuscript writing.

*Competing interests.*   The authors declare that they have no competing interests.

*Acknowledgements.*   The authors thank the many crew members, scientists, technicians, and logistics staff who have supported data collection efforts since 2008. These data would not exist if it were not for the knowledge and skill of the ships' crewmembers, most of them Greenlandic, including those of the M/Vs Viking, Adolf Jensen and R/V Tarajoq. The XCTD data were collected thanks to the patience and expertise of
the many helicopter pilots, mostly from Air Greenland. FS acknowledges the essential role played by the late Gordon Hamilton in motivating and initiating the collection of these data. We acknowledge the many scientists who contributed to this data collection. In particular, we acknowledge the contributions of D. Sutherland, R. Jackson, N. Beaird, M. Andres, D. Slater, L. Stearns, A. Lewinter, D. Finnegan and the late G. Hamilton. Invaluable technical support was provided by A. Ramsey, J. Ryder, W. Ostrom, M. Donahue and J. Kemp's entire group.

  Successfully accomplishing and funding this work has required coordination and collaboration with multiple research groups across
disciplines as we seek to answer increasingly complex and socially relevant questions. Maintaining positive collaboration with other research groups, Greenland–based scientists, logistics personnel, research vessels, and local people has been essential to ensuring observations are funded and conducted year after year.





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
