# Peer review of "A dataset for multidisciplinary applications: Thirteen years of ocean observations in Sermilik Fjord, Southeast Greenland"

_Earth System Science Data, 2025_

## Author Comment (AC1)

Response to the Reviewers

We are grateful to both Reviewers for their helpful and constructive feedback. Their reviews have greatly improved in the manuscript and added to the overall clarity of communicating the scope and importance of our work. We recognize there were distracting typos and broken LaTeX reference links in the original manuscript and appreciate the reviewers noting those.

In the following, we have written our responses to the comments where Reviewers comments are shown in *black*, and our responses in *blue*.

Updated figures based on the Reviewers suggestions are included in our comments.
* * *
**Response to Reviewer 1 (Anonymous Referee #1)**

The paper is well written and should become a useful reference for future use of this data set. There is many years of dedicated effort to collect this important dataset, and the presented summer climatology is well organized. The figures are also clear and show many examples of how the observations have been synthesized to show variability. While many of the presented results were 'as expected' for a Southeast Greenland fjord, it is good that this is published and that the 2009 – 2023 observations make sense compared to earlier work. The suggested improvements are small and few enough to tick off the 'minor' revisions here.

**General comment:** The minor revisions needed are some unclear formulations and there were also a few errors around referencing figures. The one larger issue I have is that although an impressive number of summer campaigns is presented, it is still 'summer only'. I strongly suggest that the data is described as summer data. In line 270 for example, you state; "all profiles collected in the fjord across all years". This makes it sound like you have representative data for "all years" - - you don't. You have good representation for summers. This should be corrected throughout.

We agree with this comment of clarifying that these data are summer only. It was not our intention to mislead the reader into thinking these data encompass more than the summer season. We have corrected line 270 as noted in the comment and made additional changes throughout the manuscript to provide extra clarity that these data are summer only. We note that there is one exception of a winter survey in 2010, but this is not included in the summer season climatology, and this is clarified in the text.

**Specific comments:**

Line 78,79,80: This text is repeated and should be deleted. It is also unclear what 'annual gridded dataset' is compared to the summertime means.

Repeated text has been deleted and we clarified by using "gridded summer sections" instead of 'annual gridded dataset'.

Line 100: Here you need to add "SA"; Absolute Salinity (SA) 33.3 g kg-1. Fixed

Line 157: Missing figure reference; "except for 2014 and 2020 (??)." Figure reference added and "except for 2014 and 2020" has been added for clarity.

Line 159: I don't know what you try to indicate with the (1)… Please fix. This was a broken table reference. Fixed.

Line 161: Wrong format for the deg C.  Fixed

Line 179: Probably the Greenlandic name for Midgaard Fjord is missing? Fixed

Line 180: I don't know what you try to indicate with the (2)… Please fix. This was a broken table reference. Fixed.

Line 181: Wrong format for the deg C. Fixed

Line 220: I don't know what you try to indicate with the (3)… Please fix. This was a broken table reference. Fixed.

Line 270 I suggest to just write; "all profiles collected in the fjord across all summers" Fixed.

Line 346: And here then "each summer" instead of "each year". Fixed.

Line 350: Again here you write "all the yearly data". I think that you did not include the March XCTD profiles? In any regard – please specify if this is summer data (July – September).  Fixed.

Caption Figure 4: "The 2023 CTD and XCTD data are not combined and instead treated as two separate instances of a summer fjord state leading to a total of 14 possible grids." This is clearly written also in the text, you only need to explain this once. I suggest to delete this here. Deleted.

Line 392: Here you use the correct term. You have produced "a summer state climatology".

Line 393: Missing figure reference. Fixed.

Line 418: I think you are missing something here; "The subsurface temperature minimum of this profile indicates the core of the PW layer present on the shelf." Surely – this must be the remnant of the lower part of the PW from the previous winter situation? Please clarify.

The deep, cold PW on the shelf, which we also see evidence of at the fjord mouth though it is often mixed with some fjord-origin water, is indeed indicative of winter time properties of PW and not affected by surface seasonal warming. We are less concerned here with the origins of the subsurface temperature minimum on the shelf and only make this point to highlight the PW endmember water mass on the shelf as the source of the coldest water imported into the fjord. The main point here is that the shelf PW layer is colder than the temperature minimum that we see at the fjord mouth. Perhaps the issue here is our use of the word "core" in identifying the

temperature minimum of the PW on the shelf when it is more the remnant *bottom* part of the PW layer.

We have adjusted this statement to read:

The subsurface temperature minimum of the mean profile indicates the *coldest* unmodified PW present on the shelf, on average, across all surveys from 2009--2023 and is comparable to the fjord summer climatology properties (Fig. 9). The mean East shelf *coldest* PW properties are -1.22 ± 0.35 °C and 33.17 ± 0.25 g kg\textsuperscript{-1}, occurring at a mean depth of 77 m and ranging from 50--115 m depth.

More broadly, this comment highlighted the need to create more clarity about the shelf profiles and our reason in identifying unmodified PW on the shelf. We have edited this section to improve clarity. We acknowledge that shelf profiles as part of this dataset have not been analyzed at the rigor of the fjord profiles. The shelf environment has different dynamics compared to inside the fjord. We have left this as future work and have added a statement about this in the discussion to further highlight the distinction between the fjord profiles and shelf profiles.

Figure 9 caption has missing Figure (??) reference. Fixed.

Line 449: The stratification (density) is not visible in Fig. 10. Do you mean thermocline??

Fixed. We had meant to reference Fig. 11 which shows stratification but citing the thermocline in Fig. 10 is also relevant and we have included both.

Line 463: Missing figure reference (??). Fixed.

Line 475: Figure 8 c) shows the mean salinity – not the variability. And the variability in the mouth region is only large for Temperature. Please rephrase. Fixed

Line 528: Here there is a very welcome comment on the large seasonal variability in freshwater forcing. I think you should also mention the large (trivial) solar seasonal variability that drives it. This must explain the maximum surface variability for temperature shown in Figure 8 b).

We have added this note in this section.

Otherwise it was a nice paper. The data collection is impressive and the new climatology will be very useful.

Thank you for taking the time to review our manuscript and validating the contribution of this work.

**Citation**: https://doi.org/10.5194/essd-2025-345-RC1

**Response to Reviewer 2 (Anonymous Referee #2)**

**Summary**

The authors have compiled several sources of temperature and salinity data from 2009-2023, from both CTDs and SCTDs, into what they consider to be a gridded, optimally interpolated dataset. They accounted for the differences inherent in measurements taken with different sampling tools, techniques, and missions. All of which are well accounted for in the included tables (Table 1 + 2). These collected and QCed data were then used to create along-fjord gradients, as well as summer climatologies and their root mean squares of deviation (RMSD). These data have been made available in NetCDF format, with one file for the full transect data, and individual files for the yearly collected data.

The authors claim that this dataset should provide researchers with easier to use, and more reliable data for investigations around the mean state of the fjord, it's gradient towards the mouth, and regions of variability therein. The authors provide a very thorough description of the optimal interpolation process, and investigation of the results. I enjoyed reading the history of the sampling effort given by the authors in Section 3.4. The text is very well written with a consistent and engaging voice used throughout.

However, it does not appear that the final compiled manuscript was checked as there are abundant spacing and broken reference issues in all sections. Generally, there are also quite a lot of spacing issues between symbols, numbers, and letters that the authors should triple check. Perhaps these were caused by LaTeX formatting issues? I also noted several formatting issues that appear to be notes between the authors to remember to insert certain pieces of information. These will need to be smoothed out before publication. A spelling question throughout, should 'Gletsjer' be 'Glacier'? Also, double-check that 'ice mélange' always has the accent. I'm not certain what the ESSD editorial standard is, but the authors should ensure whether a space is used for °C or not. E.g. 10°C or 10 °C. Yes, the ISO standard is 10 °C, but often in written manuscripts it is given as 10°C.

The use of "Gletsjer" was following the official naming convention of Greenland glaciers as reported in Bjørk et al. (2015). Despite being the Dutch word for glacier, it is what is considered official by the Greenlandic government and Danish entities. To avoid distraction and confusion, we substituted with English "Glacier".

We have checked that "mélange" always has the accent in the manuscript. We have double checked that the ESSD standard is 10 °C not 10°C and have made appropriate edits to comply with that in the manuscript.

Ultimately, I would argue that thes dataset presented here id not gridded data in the traditional sense of the word. Meaning, the different grid cells of data do not have evenly spaced lon/lat coordinates in any coordinate system (e.g. any satellite or model output spatial data product). I do not think that using even integers (e.g. distance from fjord mouth in km) in place of spatial coordinates can be used to call a dataset gridded. Nor are these data available at any spatial

extent, they are a linear string of data following a proscribed line, with the lon/lat coordinates attributed afterwards. I think that most readers and users, upon seeing the word 'gridded' used repeatedly throughout this manuscript would assume that the data product created here would have even pixels across the body of the fjord, with the corresponding depth values per pixel along an evenly gridded z-axis. I agree that this z-axis has been created as expected, but not the surface values. I recommend that the authors re-evaluate their use of the word 'gridded' throughout the manuscript, and rather more strongly emphasize that these are transect data, and not gridded data.

We thank the reviewer for raising this important point about terminology. We agree that "gridded data" is often used in the context of evenly spaced lon/lat coordinates covering a spatial domain, such as satellite or model output. Our dataset differs in that the gridding is applied along a prescribed fjord transect (distance from the glacier) and with depth, producing a regular section in this coordinate system (as the reviewer notes). In other words, while the data are not gridded across a 2D horizontal surface, they are interpolated onto a consistent, evenly spaced distance–depth grid along the fjord.

To address the concern and minimize confusion, we have revised the manuscript to refer to the product as *"gridded transects"* (or *"gridded along-fjord sections"*) rather than simply "gridded data." We believe this terminology preserves the accuracy of our description while clarifying the nature of the product for readers. We have also clarified this in the Methods section to distinguish our usage of "gridding" from full 2D/3D spatial gridding.

That being said, I do not have any substantive comments or suggestions before recommending this manuscript for publication. Though do see my specific comments below, particularly those on the code and data availability section. As well as my thoughts on the discussion and conclusion sections.

**Title**

- I don't know that the word 'fjord' needs to be used twice. Could just say: 'A dataset for…'.

Agreed. Fixed.

**Abstract**

- 14: 'fjord intercomparison studies' the issue with this statement is that, if this methodology has only been applied to this fjord for this dataset format, there are no other comparably formatted datasets with which one could perform fjord intercomparison studies. One may want to add a clause that datasets like the one produced by the authors would allow for effective fjord intercomparison studies, if more datasets like this one were created for other fjords. i.e. This methodology could be taken as a blue print for creating gridded temperature + salinity datasets for other fjords.

We agree with this point and stated this in the body of the paper, but we did not get into this in the abstract. We have added a statement in the abstract to make this more clear.

**Introduction**

- 81: This is a big statement. One should potentially add a sentence linking to the code or repository were this data node is beginning to develop. And/or a sentence explaining how the long-term continued support for such an endeavour has been assured.

We agree and have edited it to:

*While more coordination and work is still needed within the science community, the approach shared here is a significant step toward creating a living data repository that standardizes long--term fjord observational records into a FAIR (Findable, Accessible, Interoperable, Reusable) data format to facilitate interdisciplinary research. The proposed framework for a larger, collaborative Greenland Ice Sheet-Ocean Observing System (GrIOOS) explicitly called for a repository such as this to facilitate the use of observations of Greenland's changing environment to address socially relevant questions at local to global scales (Straneo et al., 2019).*

**Study Site**

-106: Section 2.3: It may be interesting to add some of the physical features described here (e.g. the sill) to Figure 1. As well as referencing Figure 1 somewhere in this section. Referencing Figure 1 has been fixed. We tried adding deep sill locations, but it was distracting on Fig. 1. Instead, we've edited the sentence in text to be: "The deep bathymetry of Sermilik Fjord with no shallow sills allows for direct exchange of PW and AW shelf waters (Fig. 1)." This way the bathymetry is called out and the reader is directed to Fig 1. where they can see the bathymetry.

- 115: '(GMW).In' → '(GMW). In' Fixed

- 120: Section 2.4: Why not reference Figure 1 when describing the ice mélange reason? The authors have already gone through the trouble of labelling it on the map. Yes - Fixed.

**Data**

- 157: Why the '(??)'? Referencing issue, or a note to explain the two missing years? Fixed

- 159: What does '(1)' signify? This was a latex figure referencing issue - Fixed

- 161: Throughout this and the following sub-sections, please double check the spacing between symbols, numbers, and letters. The degree symbol is incorrect. Fixed

- 179: 'GREENLANDIC NAME' ? Fixed

-180: '(2)'? Referencing issue - Fixed

- 184: Does it not cause issues to average the CTDs to 1 m depth bins, but the XCTDs to 2 m depth bins?

The XCTD were vertically-averaged to 2 m depth bins to reduce noise that is inherent in the XCTD measurement. The XCTD profiles are noisier than the CTD profiles and only trustable at 2 m increments. This is based on a review of properties and comparison with CTD profiles, and why we made the 2 m vertically averaged XCTD profiles publicly available. For our purposes here, the 2 m binned XCTD profiles were then resampled to 1 m to integrate with the CTD profiles and move forward in this analysis. We have added this statement for clarity in the manuscript.

- 220: '(3)'? I take it that these are meant to be references to the Figures? How are references to tables made? 'Figure 1' is referenced as such in Section 4. Reference Issue - Fixed

 **Methods**

- 245-251: It seems sub-optimal to me that the grids would not be consistent for both temperature and salinity, nor the same for every year. If indeed one of the objectives of the creation of this dataset is to make it useful to a wide range of disciplines, the grids should be static (i.e. the same for each variable and year). Or, it seems that the use of the word 'grid' in this text is not what I would expect it to be, and I've misunderstood the description of the methodology. Rather, the authors use the word 'grid' to refer to the mathematical equation used to calculate the average values per grid cell? With a grid cell being the used in the traditional sense, *a la* gridded data found within NetCDF files. Again though after reading the full manuscript I see that this is not the case either. I have put my thoughts on this topic in the summary above.

The temperature and salinity grids are consistent with each other and have the same vertical coordinates (m depth) and horizontal coordinations (km distance from glacier). However, the *extent* of the grids varies based on the maximum extent of the observations ie. we do not map beyond where there are observations in the domain). We believe this is reasonable. Figure 4 shows the different extents of each section to make this clear, but they all have the same gridded coordinates.

The confusion lies in the sentences: "This process was performed independently for temperature and salinity variables. The depth extent of the objectively mapped grid for each summer season was determined by the depth of the deepest profile and the horizontal extent was determined by the minimum and maximum along–fjord distances of the profile locations for each survey. This results in different extent grids bounded by the maximum observational extent for each summer."

For clarity we have edited this section to: "This process was performed independently for temperature and salinity variables. While each section is ultimately mapped to the same grid cell locations, the data extent of each gridded section varies because they are bounded by the minimum and maximum along–fjord distance locations of the profiles in each survey . The deepest vertical extent of each gridded section is bounded by the maximum depth profile for each survey (Fig. 4).

We have addressed the other concerns about using the word "grid" above comments.

- 289: '(Eq. 4 and Eq. 4).' -> '(Eq. 4 and Eq. 5).' Fixed

- 343: 'The 2023…' This seems like it would cause issues for users that simply want to extract annual climatology values. Ah, after accessing the .nc data files I see that the authors have chosen to give each year as an individual NetCDF file. Thereby avoiding the issue of the two different gridded datasets for 2023. This is a sub-optimal way to provide data for non-technical users as they are not necessarily going to have the competence to access and extract data well from multiple disparate files. I advise that the authors combine all annual data into one NetCDF file. Keeping the 2023XCTD data as a supplementary file.

We already have included all the gridded sections in a single file in SermilikGriddedSections_v1.nc as this suggests. The individual year section files (eg. SermilikSection_2009.nc) are the ungridded input data of selected discrete profiles so that users can see the input data if needed and can also run the provided code with this input data. In response to this confusion, we have updated the metadata descriptions for improved clarity on the Arctic Data Center and GitHub repositories.

**Results**

- 367: Spacing issues throughout the paragraph. Fixed

- 379: 'mapping relative' -> 'mapping of relative'

The variable being calculated here is termed "mapping relative error" and also referred to as "mapping error".

- 379: 'The user...' This is a nice idea, but outside of physical oceanography, I don't think many research teams are honestly going to get into this level of detail. I make this point considering that the authors hope for this dataset to serve as the locus of development for an established community resource. The more simple a product is to use, the more likely people are to use it. The more people use it, the more likely it will be funded in the future. Too much complexity may be counter-productive.

While we agree that it is a deep level of detail, this is a specific benefit of the objective mapping method. Grid cells with larger magnitude errors can be masked out if needed. We have included a sentence after this stating: *"The mapping relative error magnitudes of all sections are acceptable for most use cases as shared here".*

- 389: 'Supplementary Material' -> 'Supplementary Material (Figures S1-S8)' Fixed

- 393: Broken figure reference. Fixed

- 463: Broken reference. Fixed

**Discussion**

- I find the discussion a bit odd in that, so much of the data were analysed and reported on in the results, but these are not discussed in this section. Rather the authors decided to give three considerations on the use of these data. While good to include, would it not make sense to include a sub-section discussing any of the interesting results that were made possible due to the data interpolation method used here?

We understand this point of view and there are differing opinions about the role of a discussion section vs results section. Our intent was indeed to present the key physical features and patterns in the Results, so that the Discussion remains focused on issues of usability, caveats, and future applications of the dataset. However, we accept that a more explicit connecting bridge between Results and Discussion—highlighting how the method led to key and novel results—would improve clarity and help readers see the value added by the gridded transect product.

In response, we have added a new sub-section at the start of the Discussion (Section 6.1) entitled "Summary of results facilitated by gridded sections", in which we highlight the result takeaways (which have more in depth context within the previous Result subsections). This does not duplicate the full Results text but places them in the context of why the gridded section products are useful and increase our knowledge of the system beyond discrete profiles.

We believe this addition responds to the reviewer's point while maintaining the focus of the Discussion on the dataset's utility, limitations, and future applications.

**Conclusions**

- 571: 'interdisciplinary disciplinary' → 'interdisciplinary' Fixed

- 572: 'shared' → 'created' Fixed

- I find that this section also does not match with the body of the results. The authors have put quite a lot of efforts into fastidiously documenting the resulting data of their applied interpolation method, but then hardly discuss or conclude anything from this. Rather they choose to focus on the importance of FAIR data to the Arctic research community. I completely agree that this is important, therefore, it seems that throughout the results section (or discussion) the results that describe the data (e.g. mean inner-fjord deep temperature) should be provided with some context for how this is relevant to FAIR data in the Arctic.

We thank the reviewer for this suggestion. In response, we have revised the Conclusion to more clearly summarize and emphasize the key results enabled by the gridded sections. The FAIR data discussion is now framed as a natural extension of these contributions and the flexibility provided by the data format and results, ensuring the Conclusion aligns with both the Results and the broader utility of the dataset.

**Code and data availability**

- Link given for datasets work as expected. Data can be downloaded and are in a standard NetCDF format.

- The code used for these analyses is also publicly available. In the same location as the data themselves, which is ideal.

- Upon loading the transect (gridded) data, I do not have the same smooth visual summer climatology shown in the results of the manuscript. Regardless of which x and y axis variables are used.

We think something must be amiss here. The climatology gridded fields are contained in SermilikGriddedClimatology_v1.nc. This file is complete and plotting the temperature and salinity and associated RMSD fields will recreate all panels in Fig. 8.

SermilikGriddedSections_v1.nc includes the individual gridded transects for each summer survey in one NetCDF file. Plotting one of these fields will only reveal an individual summer's gridded field, not the climatology.

Finally, also included SermilikSection_XXXX.nc (where XXXX = each year. eg. *SermilikSection_2021.nc*). These data are NOT gridded fields but are the selected discrete profiles used as the input data used in the gridding method. These files also include the residuals at each profile location showing the comparison between the values in the gridded fields and original profile values.

We think that perhaps the reviewer was not plotting the climatology, but plotting an individual survey either gridded (from SermilikGriddedSections_v1.nc) or the original, discrete profiles (from SermilikSection_YEAR.nc)? However, due to this confusion, we have updated both the Arctic Data Center and GitHub pages to be more clear about the role of each file and what data is contained in each.

- If the authors are serious about this methodology being able to be easily applied by other teams to create similar datasets, it would be ideal to translate it into Python and R code. *I am not recommending this be done for this manuscript.* Rather something to add to a long-term TO DO list. We appreciate this suggestion and also agree!

**Table 1**

Very nice. However, this table is not referred to in the text.

Missing reference link. Fixed.

**Table 2**

Also nice, but also not referred to in the text.

Missing reference link. Fixed.

**Table 3**

No comments.

**Figure 1**

If the authors want to show depth contours, it would be better to use an illustrated figure (e.g. Figure 2), rather than a satellite image. By showing the water as dark blue (true colour), this figure gives the impression that the majority of the fjord is deeper than 800 metres. While the ice mélange regions appear to be above sea level. However, I agree that it is useful to illustrate where the ice mélange region is, so I would rather just remove the depth contours as they are shown in Figure 2, and showing the depth is not the primary purpose of this figure.

We argue that showing depth is important in this figure to acquaint readers with important features of the region. We agree though that there is a balance to strike in using the satellite image to show mélange extent and also show depth contours. We have addressed this by removing the depth contours in the mélange region. The bathymetry data in this region are likely highly interpolated or from model results in any case because direct bathymetry observations here are very scarce.

[Figure]

**Figure 2**

Why is this figure not referred to in the text? One should show the x- and y-axis labels only once for the entire figure. Not once for every facet. This would allow for the labels to be a larger, more legible, font size. I like the authors choice to but the legend in the top left, and to start the first facet in the top middle of the figure. Labels have been fixed as suggested.

[Figure]

**Figure 3**

A very useful figure. A good example of one of the few cases when it is correct to use a stacked barplot.

**Figure 4**

Well presented and clear. This figure helped me to realise I was misunderstanding what the authors meant when using the word 'grid' vs 'grid cell'.

**Figure 5**

The residuals (b, e), shown as dot plots, are not easy to read. Though I don't have a better suggestion. Upon zooming into the PDF I was able to make out the values relatively well. The choice of a green-purple colour scale is a good one. Otherwise a very well assembled figure. The authors may want to consider increasing the size of all numbers, labels, and text. Though I leave this to the editorial process. As with Figure 1, it could save space and text on the figure to give the x- and y-axis labels only once as they are always the same.

**Figure 6**

Another nice figure. I don't think it's necessary to state what the bin size is along the x-axis. An interested reader could determine that for themselves.

**Figure 7**

There is a lot going on in this figure. The caption does a good job of describing how to interpret the information, and it is interesting to look at the difference between the original profiles and the gridded data.

**Figure 8**

Another consistently good figure.

**Figure 9**

I like the choice the authors took to inset the horizontal profile panel in the upper left corner. There is a broken figure reference in the caption. Fixed

**Figure 10**

Considering that a light colour palette is being used, it may be better to make the lines slightly thicker. Or highlight the edges of the lines in black. Though that can lead to other visual artefacts. We understand this note, but it's a balance between not losing the black mean profile for each panel against the other lines. This is why we chose the lighter colored palette. We tried thickening the lines more, but this looks heavy and loses some of the structure of each line.

**Figure 11**

Same comment as Figure 10. 'larger the' 'larger than the'. Fixed.

**Figure S1-S8**

The same comments for their corresponding versions in the main manuscript. I think it was a good choice by the authors to have included all of the profiles in the supplement like this.

**Citation**: https://doi.org/10.5194/essd-2025-345-RC2
Thank you for taking the time to carefully and thoughtfully review our manuscript and validating the contribution of this work.

---

## Author Comment (AC2)

[revised manuscript text omitted]

 While more coordination and work is still needed within the science community, the approach shared here is a significant step toward creating a  living data repository that standardizes long–
85    term fjord observational records into a FAIR (Findable, Accessible, Interoperable, Reusable) data format  to facilitate interdisciplinary research. The proposed framework for a larger, collaborative Greenland Ice Sheet-Ocean Observing System (GrIOOS) explicitly called for a repository such as this to facilitate the use of observations of Greenland's changing environment to address socially relevant questions at local to global scales (Straneo et al., 2019).

[revised manuscript text omitted]

**4 Methods**

**4.1 Profile selection for analysis**

 Prior to the gridding process, we constructed along–fjord sections for each  survey following the thalweg transect line (Fig. 1). This required careful manual selection of  profiles located nearest to the thalweg  line and capturing similar fjord conditions within a certain time period. If profiles in similar locations existed, but were collected at different time periods during the survey (eg. collected while sailing upfjord and then downfjord several days later), we only retained the profile that  created the best continuous synoptic section. If multiple profiles were collected in the across–fjord direction at the same along–fjord distance then those across-fjord profiles were averaged and the mean profile was used in the along–fjord section. Profiles near the shallower (< 300 m) fjord sides were not included. The final selection of individual profiles and mean profiles (averaged in the across-fjord direction) making

up the best synoptic along–fjord hydrographic section were then used as input data to create the gridded along–fjord dataset.

Data from inside the plume regions from years 2016 and 2019 are not included in the along–fjord sections. This is because the gradients of properties are at a finer scale in this dynamic region than we are accounting for in the objective mapping process. These plume data are discussed separately. Similarly, dynamics in the shelf region occur at different scales compared to the fjord and we do not use the shelf profiles for constructing the fjord sections.

Prior to objective mapping, we perform a "bottom fill" procedure for profiles extending beyond 550 m or deeper to enhance data density of the deep fjord regions for the objective mapping process. For all profile locations, properties below the sill depth of 550 m show little variability and are remarkably stable with respect to depth, but they do vary in the along–fjord distance. Without bottom filling of these profiles, the deepest profile informs the properties at that depth across the fjord when it is more likely that properties are similar to their nearest vertical neighbors. First, for all profiles extending 550 m or deeper, we calculated the average temperature and salinity value of the deepest 10 m of that profile. Then, we extrapolated these properties uniformly to the bottom. This extrapolation procedure was used for 128 profiles out of a total of 172 used in the along–fjord sections.

**4.2 Creating gridded data using objective mapping**

The challenge of creating gridded fields from scattered observations is well known in the earth sciences and there are many possible approaches. Objective mapping (also referred to as optimal or optimum interpolation) allows for the explicit use of input parameters and use of multiple spatial correlation scales to better represent physical processes. Objective mapping approaches are commonly applied to other hydrographic profile datasets including from the northern Antarctic Peninsula (Dotto et al., 2021) and the Weddell Sea (Reeve et al., 2016), and of biogeochemical profiles in the Southern Ocean (Mazloff et al., 2023). These previous applications are concerned with larger ocean basin–scale observations, often involving thousands of profiles, and spanning decades. This is the first application of objective mapping for a Greenland fjord. Only recently have we been monitoring Sermilik Fjord long enough (> 10 summer seasons) and with dense enough observations to appropriately inform the parameters and assumptions of the interpolation method. The increased utility provided by a gridded section dataset became apparent as research about Greenland fjords is maturing and data volume is increasing. We note that our use of "gridded data" here refers to gridded hydrographic sections (or transect) using a 2D coordinate system consisting of an along-fjord horizontal direction and depth. This is in contrast to other forms of gridded data with a 3D coordinate system (eg. outputs of regional ocean models using latitude, longitude, and depth levels) or a map view 2D coordinate system of latitude and longitude (eg. satellite data).

The along–fjord sections constructed with the final selected discrete profiles for each year were mapped onto a 2 km (horizontal) x 5 m (vertical) grid with the objective mapping procedure.  The horizontal along-fjord coordinate system is referenced using 0 km at the 2019 location of Helheim Glacier's terminus and follows the thalweg section line. Associated latitude and longitude coordinates of gridcells this transect are included in the data products. The gridding process was performed independently for temperature and salinity variables.

 While each section is ultimately mapped to the same grid cell locations, the data extent of each gridded section varies because they are bounded by the minimum and maximum  along–fjord distance locations of the profiles in each survey. The deepest vertical extent of each gridded section is bounded by the maximum  depth profile for each survey (Fig. 4).

[revised manuscript text omitted]

In the upper water column, there is a subsurface temperature minimum (50 m–100 m, $\sigma_0$ =  26.5 kg m$^{-3}$)
445 below a near–surface warm layer. This subsurface temperature minimum is at a similar depth and density  to the cold, unmodified PW layer typically observed on the shelf (Sutherland and Pickart, 2008). However, the  fjord mouth subsurface temperature minimum is 1  C warmer than the established properties of unmodified shelf PW (< 0 °C) .
450

To better interpret the properties at the fjord mouth, we identified unmodified PW properties from shelf profiles included in this dataset. Shelf profiles were collected during each survey, but are not included in the creation of the gridded data for the fjord. We created a single representative shelf profile for each survey by taking the temperature minimum of each isopycnal band across all  East section shelf profiles, which sample the inflowing shelf waters, for a given survey (Fig. 7).
455 We then calculated a mean profile from  all representative East section shelf profiles from each survey. The subsurface temperature minimum of  the mean profile indicates the  coldest unmodified PW present on the shelf. , on average, across all surveys from 2009–2023 and is comparable to the fjord summer climatology properties (Fig. 9). The mean East shelf coldest PW properties are -1.22 $\pm$ 0.35 °C and 33.17 $\pm$ 0.25 g kg$^{-1}$, occurring at a mean depth of 77 m and ranging from
460 50–115 m depth.

 These properties agree with previous observations of unmodified PW on the shelf outside of Sermilik Fjord (Sutherland and Pickart, 2008; Harden et al., 201 . However, the properties we calculated here are representative of more surveys allowing for improved understanding of the
465 stability and variability of PW properties, and their influence on fjord properties. From this brief analysis, we can conclude that, on average, the temperature minimum at the fjord mouth in the 50–100 m layer represents a mixture of PW and fjord-origin GMW that is nearly 1.5 °C  warmer than unmodified shelf PW.

[revised manuscript text omitted]

525 shelf PW for those surveys. In other years (2010, 2018, 2019, 2020), more representative of the average, shelf PW properties are not present in the fjord and the fjord mouth temperatures are much warmer than shelf PW. Many of the mouth profiles show characteristic interleaving patterns above 200 m, where shelf and fjord GMW waters of similar density are meeting and mixing.  Note that the average mouth profile is less smooth compared to the average near glacier profile due

530 to multiple intrusions with sharp thermoclines at various depths being averaged together. This also impacts the stratification, which shows the largest range of yearly values between 50–200 m at the mouth (Fig. 11).

**5.3.2 Near glacier variability**

In the near glacier region, the upper 50 m has the smallest temperature RMSD of the fjord domain. This highlights the consistency of the ice mélange in setting the temperature properties in this area. From  50–100 m, there is an increase

535 in the temperature RMSD (maximum ± 1.09 °C) as the thermocline is more variable– likely in response to variability in GMW extent in the water column and properties. Below the thermocline, the average RMSD of the GMW layer properties between  165–300 m are ±0.51 °C and ±0.14 g kg$^{-1}$. As previous studies have noted, the properties of this GMW layer vary with the unmodified AW properties at depth near the glacier as these are the waters being upwelled (Muilwijk et al., 2022). When AW near the glacier is cooler (warmer) than average for a given year, the GMW properties resulting from upwelling in

540 the SGD plume are cooler (warmer) than average.

**5.3.3 Mid–fjord variability**

Profiles from the mid–fjord region for each  survey show similarly variable interleaving and intrusion features as the mouth profiles, though the mid–fjord intrusions are less sharp overall. Some years, properties in the mid–fjord region are more similar to an average near glacier profile (2015, 2017, 2023) with < 0 °C temperatures in the upper 50 m and exhibiting a strong

545 thermocline between  50–100 m and relatively weak intrusions. Other years are more similar to a mouth profile (2011, 2012, 2021) with warmer surface waters and multiple stronger intrusions. The along–fjord location of the transition between near glacier properties and mouth properties varies from year to year. This variability is not captured in the time mean mid–fjord profile and climatology of the mid–fjord region.

[Figure]

**Figure 11.** $N^2$ (Brunt–Vaisala frequency), a measure of stratification, from the near glacier (a), mid–fjord (b), and mouth (c) regions for all yearly gridded data. The profiles are from the same locations as in Figure 10. The upper panels have a different vertical scale to emphasize the surface. The horizontal scale in the upper panels is an order of magnitude larger than the lower panels. $N^2$ was calculated from conservative temperature and absolute salinity gridded data using the TEOS–10 Oceanographic Toolbox (McDougall and Barker, 2011). Black bold lines represent the mean $N^2$ profile for all years in each region.

**5.3.4 Yearly anomalies from the mean**

550    To further investigate the temporal variability, we subtracted the climatology from each  summer grid to produce the temperature and salinity anomaly for each  grid (selected years shown in Fig. 12). The temperature anomaly fields show  summers where nearly the entire fjord domain is warmer (e.g. 2019) or colder (e.g. 2015) than average. 2021 is the year with the smallest average anomaly across the whole domain, however, there are still areas in the domain that are warmer and colder than the summer average.

[Figure]

**Figure 12.** Conservative temperature anomalies from the climatological mean for yearly grids 2009 (a), 2015 (b), 2019 (c), and 2021 (d). Cooler (warmer) colors show values less (greater) than the mean for each grid cell.

555      The pattern in the gridded anomaly field can be both due to variability in the properties and variability in the depth of the thermocline. For example, the properties of the temperature minimum at the mouth could match the mean temperature minimum properties, but if the thermocline is at a different depth the anomaly will be nonzero.

**6   Discussion**

560   **6.1   Summary of results facilitated by gridded sections**

The method presented here, and the creation of gridded sections from discrete, irregular profiles, enables novel insights and quantification of water properties in Sermilik Fjord. Most notably, the summer climatology product, along with complementary information on the range and patterns of variability, is only possible with regularly gridded sections and could not have been constructed from the original profiles alone. The same is true for the calculation of the anomalies relative to the climatological

565   state, which allow direct interannual comparisons against a mean field. For example, our analysis shows that 2015 was an anomalously cold year in the fjord, suggesting that conclusions from previous studies based solely on 2015 data should be interpreted with caution.

Beyond climatology and anomalies, the gridded sections also reveal coherent fjord-scale property structures that are less evident in scattered profile data. For instance, variability in Atlantic Water along the fjord is more readily discerned in the gridded fields than in the raw profiles. The gridded format also enables straightforward calculation of spatially averaged water properties in different parts of the fjord, both for individual surveys and for the summer climatology. These mean properties, presented in the Results, are robust quantities that can be directly used in modeling and comparative studies. Together, these examples demonstrate that the gridded sections are not just improving the accessibility of the data, but also the interpretive power, producing results that would not otherwise be attainable.

[revised manuscript text omitted]

While we have not included the plume polynya profiles in the gridded products, the individual profiles are available in the original data and these can be useful in studies employing plume models. Similarly, the shelf profiles were not included in the gridded data presented here as the shelf environment dominated by different dynamics, but the individual profiles are available

for use and provide important context for interpreting fjord water properties. We chose to create an average East section shelf profile to aid in the interpretation of properties at the fjord mouth. A more rigorous analysis of shelf water mass properties and the creation of a shelf summer climatology is possible with this dataset and will be considered in future work.

**7**

The dataset and gridded products presented in this study provide a crucial step toward standardizing and centralizing long–term fjord observations in Greenland. By compiling 13 years of hydrographic data from Sermilik Fjord, we offer a comprehensive and accessible resource for studying fjord dynamics and ice–ocean interactions. The combined CTD and XCTD observations lead to greater spatial coverage of the fjord, including the  mélange region for multiple years and the subglacial discharge plume polyna region for two years. The objective mapping method  enabled necessary and novel analyses that are not possible from the raw profiles alone, including the construction of an along-fjord summer climatology, quantification of interannual anomalies, and identification of water property features. Importantly, these results provide context for interpreting previous  work in Sermilik Fjord. We demonstrated how other quantities (eg. $N^2$) and water properties of specific regions can easily be calculated from the  gridded sections depending on questions of interest. Finally, the method used to generate gridded sections is adaptable for different variables and fjord settings and can facilitate interdisciplinary research– enabling comparisons with models, biological data, and other observations.

This work highlights the need for a coordinated approach to fjord data collection and sharing. Establishing a structured, FAIR–compliant data repository for Greenland fjords will improve the accessibility and utility of these critical datasets, ultimately enhancing our understanding of glacial fjord systems and strengthening collaboration within the international science community and with Greenlandic partners.

**8**

The gridded data products for each individual year and the climatology are available at the Arctic Data Center (https://doi.org/10.18739/A2513TZ0P, Roth et al. 2025) and GitHub (https://github.com/a1roth/sermilik_gridded_hydrography) as netCDF files. The files contain gridded section of $\Theta$ and $S_A$, as well as their respective mapping relative error matrices, for every summer survey. Derived variables, like potential density or $N^2$, can be calculated by the user. The thalweg along–fjord gridded coordinates in distance (km) from the 2019 Helheim Glacier terminus position and in latitude and longitude coordinates are included in the files.

[revised manuscript text omitted]